# Kinetic mass-transfer calculation of water isotope fractionation due to cloud microphysics in a regional meteorological model

I-Chun Tsai[1], Wan-Yu Chen[2,3], Jen-Ping Chen*[2,4], and Mao-Chang Liang[5]

1. Research Center for Environmental Changes, Academia Sinica, Taipei, Taiwan R. O. C.

2. Department of Atmospheric Sciences, National Taiwan University, Taipei, Taiwan, R.O.C.

3. Central Weather Bureau, Taipei, Taiwan, R.O.C.

4. International Degree Program on Climate Change and Sustainable Development, National Taiwan University, Taipei, Taiwan, R.O.C.

5. Institute of Earth Sciences, Academia Sinica, Taipei, Taiwan R. O. C.

Draft submitted to Atmospheric Chemistry and Physics (ACP)

* Corresponding Author:

Jen-Ping Chen, Professor

Department of Atmospheric Sciences

National Taiwan University

No. 1, Sect. 4, Roosevelt Road, Taipei, Taiwan 10673

Email: jpchen@ntu.edu.tw

Phone: +886-2-33663912Fax: +886-2-23633317

1                                    Abstract

2          In conventional atmospheric models, isotope exchange between liquid, gas and

solid phases is usually assumed to be in equilibrium, and the highly kinetic phase
transformation processes inferred in clouds are yet to be fully investigated. In this study,
a two-moment microphysical scheme in the NCAR Weather Research and Forecasting
(WRF) model was modified to allow kinetic calculation of isotope fractionation due to
various cloud microphysical phase-change processes. A case of moving cold front is
selected for quantifying the effect of different factors controlling isotopic composition,
including water vapor sources, atmospheric transport, phase transition pathways of
water in clouds, and kinetic versus equilibrium mass transfer. A base-run simulation
was able to reproduce the ~50‰ decrease in $\delta D$ that observed during the frontal
passage. Sensitivity tests suggest that all the above factors contributed significantly to
the variations in isotope composition. The thermal equilibrium assumption commonly
used in earlier studies may cause an overestimate of mean vapor-phase $\delta D$ by 11‰,
and the maximum difference can be more than 20‰. Using initial vertical distribution
and lower boundary conditions of water stable isotopes from satellite data are critical
to obtain successful isotope simulations, without which the $\delta D$ in water vapor can be
off by about 34 and 28‰, respectively.    Without microphysical fractionation, the $\delta D$
in water vapor can be off by about 25‰.

## 1 Introduction

The water stable isotopes ($^{1}H_2O$, $^{1}H^{2}D^{16}O$ and $^{1}H_2^{18}O$) differ in molecular symmetry and weight. These differences in physical properties lead to a change in the stable isotope composition of water, due to fractionation during phase changes. When water vapor condenses and forms liquid or solid particles, it becomes depleted in $^{2}D$ and $^{18}O$, because heavy isotopes condense preferentially to light ones. Information about the stable water stable isotopes is thus useful for understanding the water cycle (Dansgaard, 1964; Dawson and Ehleringer, 1998; Lorius et al., 1985; Risi et al., 2012; Sturm et al., 2010).

Isotope fractionation, as measured in precipitation, has been studied for decades. The observed isotope concentrations generally exhibit significant variations in either time or space. Factors such as surface type (e.g., land versus ocean), latitude, temperature, and precipitation amount effects are commonly considered to be key to the relationship between isotope fractionation and meteorological parameters (Dansgaard, 1964; Gonfiantini, 1985; Rozanski et al., 1993; Yurtsever and Gat, 1981; Kurita, 2013; Zwart et al., 2018). These factors are related to various physical processes, such as the surface water vapor source, atmospheric transport, phase changes in clouds and gravitational sorting of precipitation hydrometeors. For example, the water stable isotopic ratios decreased inland from the coast and the so-called continental effect (Clark and Fritz, 1997). The precipitation amount effect states that isotopic contents of tropical precipitation decrease as the amount of local precipitation increases (Dansgaard, 1964; Kurita, 2013), and the cause of which could be either the preferential removal during condensation (Cole et al., 1999; Yoshimura et al., 2003) or stronger downdraft in more intense convection (Risi et al., 2008). Untangling the intertwined effects of the various physical processes is essential to understanding isotope fractionation and the atmospheric water cycle.

The variations in isotope concentrations usually have multiple causes, and it is

difficult to understand the impacts of different factors by measurements alone.
Therefore, numerical models have been used to simulate isotope fractionation in the
atmosphere. The Rayleigh-type models, in which the air mass is continuously cooled
down and the condensation process is assumed to occur in isotopic equilibrium, are
widely used in discussing isotope measurements (Aldaz and Deutsch, 1967; Dansgaard,
1964). Such models can explain the linear relationship between the surface
temperature and isotopic composition of precipitation (Rozanski et al., 1993), and have
been expanded to incorporate more processes since the publication of Dansgaard
(1964). For example, Jouzel and Merlivat (1984) reported that the isotopic equilibrium
assumption led to an overestimation of the temperature-isotope gradients of polar
snow, so they included isotopic kinetic effects at snow formation in the models.
However, the Rayleigh-type models greatly simplify the complexity of the hydrological
cycle, and Joussaume et al. (1984) introduced the concept of building isotopes into an
atmospheric general circulation model (AGCM). AGCMs can calculate the transport
and mixing of air masses from different sources (which cannot be addressed by the
Rayleigh-type models), and have been used in studying the hydrological cycle in the
troposphere (Hoffmann et al., 1998; Lee et al., 2007; Sjolte and Hoffmann, 2014;
Yoshimura et al., 2008). In conventional AGCMs, isotope exchange between liquid or
ice and gas phases is usually assumed to be in a partial or full equilibrium state
(Hoffmann et al., 1998, Risi et al., 2010, Nusbaumer et al., 2017, Werner et al., 2011,
Yoshimura et al., 2010). In a synoptic weather system such as a front or typhoon,
thermal equilibrium fractionation may not be appropriate for describing fractionation
during phase change since the clouds are usually not in vapor equilibrium (Laskar et
al., 2014). Therefore, in recent years, several regional models start to consider the
kinetic fractionation during evaporation from open water, condensation from vapor
to ice, or isotope exchange from raindrops to unsaturated air (Hoffmann et al., 1998;
Yoshimura et al., 2010; Blossey et al. 2010; Pfahl et al., 2012; Dütsch et al., 2016).
However, the microphysics in these global or regional models are usually described
with single moment schemes. This study developed a kinetic fractionation scheme for
water stable isotopes using a two-moment microphysical scheme that coupled into the
National Center for Atmospheric Research (NCAR) Weather Research and Forecasting
(WRF) model (Skamarock, 2008) To understand the role of different factors in the
fractionation of the stable isotopes of water at the synoptic scale.

Because the $\alpha_{l-v}$ of $^{18}O$ (grey line in Fig. 2) does not deviate significantly from

unity, so the signal of $^{18}O$ fractionation is generally much less pronounced. Therefore,
we focus on deuterium for demonstrating the fractionation processes. The
microphysical processes of deuterium such as condensation and collision were
incorporated into WRF. A moving frontal system is selected to demonstrate the effect
of microphysical fractionation versus other controlling factors such as air mass origins
and surface sources. The effects of microphysical processes, including kinetic versus
equilibrium treatments, are discussed in more details; whereas the importance of
initial and boundary conditions of vapor-phase isotope is also investigated.

2     Methodology

In this study, the WRF model version 3.4.1 coupled with a two-moment bulkwater

microphysical scheme (cf. Cheng et al., 2010; Chen et al., 2015; Dearden et al., 2016)
that was developed at the National Taiwan University (hereafter, the NTU scheme) was
selected for simulations. The NTU scheme shown in Fig. 1 is modified to handle the
isotope fractionation due to various cloud microphysical phase-change processes.
The HDO cycle and their initial and boundary conditions were incorporated into the
model and more details were provided in section 2.1. The simulation setup and
observation data are given in section 2.2 and 2.3, respectively.

2.1  Description of the isotopic microphysical model

In modified NTU scheme, isotope mass transfer between vapor-, liquid-, and ice-

phase hydrometeors during microphysical processes such as deposition, sublimation,
evaporation, and condensation, were considered explicitly (cf. Fig. 1).    For processes
of collision-collection or melting/freezing, the masses of isotopes of the involving
particles are simply combined or conserved, respectively, without worrying about the
fractionation.

Thermal equilibrium fractionation has been widely used in conventional models.

In such schemes, the HDO concentration can be determined from the $H_2^{16}O$ (hereafter,
$H_2O$) concentration for both gas and liquid phases, because it is assumed that HDO is
always in equilibrium with $H_2O$, irrespective of their phase states. The equilibrium
between stable isotopes in liquid water and vapor phases is commonly expressed using
the isotopic fractionation factor $\alpha_{l-v}$:

$$\alpha_{l-v} \equiv \frac{R_l}{R_v} \tag{1}$$

where $R$ is the ratio of the heavy (HDO) to light ($H_2O$) isotopes. This ratio can be
explained with the Raoult's law, which states that the activity (saturation ratio) of each
species in the vapor phase equals its activity in the liquid phase. For the HDO–$H_2O$
system, this relationship can be expressed as:

$$\frac{n_{HDO}}{n_{HDO}+n_{H2O}+n_x} = \frac{P_{HDO}}{P_{s,HDO}} \tag{2a}$$

$$\frac{n_{H2O}}{n_{HDO}+n_{H2O}+n_x} = \frac{P_{H2O}}{P_{s,H2O}} \tag{2b}$$

where $n$ is the number of moles in the liquid phase, $P$ is vapor pressure, $P_s$ is
saturation vapor pressure, whereas $x$ represents all other chemical species. By
dividing (2a) by (2b), one can derive the following:
$$\frac{\frac{n_{HDO}}{n_{H2O}}}{\frac{P_{HDO}}{P_{H2O}}} = \frac{P_{s,H2O}}{P_{s,HDO}}$$
(3)

One can see that the left-hand-side term is exactly $\alpha_{l-v}$, while the right-hand-side
term tells us that this factor is actually the ratio between the saturation vapor pressure
of $H_2O$ and HDO. Thus the isotopic fractionation factor $\alpha_{l-v}$ is a function of
temperature only, and can be determined experimentally. In this study, we adopted
the temperature dependence of $\alpha_{l-v}$ from Horita and Wesolowski (1994):
$$10^3 \cdot ln\alpha_{l-v} = 1158\left(\frac{T^3}{10^9}\right) - 1620.1\left(\frac{T^2}{10^6}\right) + 794.84\left(\frac{T}{10^3}\right) - 161.04 + 2.9992(\frac{10^9}{T^3})$$
(4a)


whereas that between ice and water vapor was adapted from Ellehoj et al. (2013).
$$ln\alpha_{s-v} = ln\frac{R_S}{R_v} = 0.2133 - \frac{203.10}{T} + \frac{48888}{T^2}$$
(4b)

where the subscript "s" means solid phase.
When kinetic process is considered, isotopic fractionation is not only related to
temperature but also factors such as the diffusion coefficient and water vapor
concentration. The calculation of kinetic fractionation during
condensation/evaporation is based on the two-stream Maxwellian kinetic equation:
$$\frac{dm_{HDO}}{dt} = 4\pi r D_{HDO}\left(\rho_{env,HDO} - \rho_{p,HDO}\right)$$
(5)

where $m$ is HDO mass in the particle, $t$ is time, $r$ is hydrometeor particle size, $D$
is the mass diffusivity in air, $\rho_{env}$ is vapor density in the air, and $\rho_p$ is vapor density
at the particle surface. The latter two terms can be rewritten as:
$$\rho_{env,HDO} = \frac{P_{HDO}}{R_{HDO}T_{air}} \quad \text{and} \quad \rho_{p,HDO} = a_{HDO}\frac{P_{s,HDO}}{R_{HDO}T_p}$$
(6)

where $R_{HDO}$ is the gas constant of HDO; $a_{HDO}$ and $P_{s,HDO}$ are the activity and
saturation vapor pressure of HDO, respectively; and $T_{air}$ and $T_p$ are temperatures
of air and particle surface, respectively.   Equation (5) is for single particle, but the
bulkwater microphysical schemes commonly used in regional weather models deal
with a population of hydrometeor particles (thus called bulkwater). Conventional
bulkwater schemes apply a mathematical function to represent the size distribution of
any hydrometeor category, and the matnematical function is solve by knowing several
bulk properties (moments) of the size distribution. The NTU scheme is a two-moment
scheme that predicts both the number and mass concentrations of each bulkwater
category, which allows better presentation of microphysical processes than the
commonly used one-moment schemes (Taufour et al., 2018). In contrast to the
conventional bulkwater schemes that must assume a certain size distribution function,
the NTU scheme derived the warm-cloud parameterization by analyzing results from
bin model simulations and thus is rather accurate and comprehensive in microphysical
processes; while the cold-cloud parameterization still follows the conventional
approach. Another advantages of the NTU scheme is that it does not apply the
"saturation adjustment" strategy, as done in most global and regional models. This
saturation adjustment treatment assumes that water vapor and liquid (or ice) water
are in thermodynamic equilibrium once water (or ice) saturation is reached in non-
mixed-phase clouds (i.e., all hydrometeors are either liquid or ice). Therefore, for
models applying the saturation adjustment strategy, condensation is not calculated
explicitly but rather by converting all excess water vapor into condensate regardless of
the cloud drop size and number concentration or the time needed for condensing out
all supersaturated water. So, under the saturation adjustment assumption, kinetic
effect as described in Eq. (5) cannot be solved fully and explicitly. In mixed-phase
clouds (i.e., water and ice coexist), the equilibrium is maintained by assuming either
water saturation or ice saturation (e.g., Sundqvist, 1978), or by varying linearly from
water saturation to ice saturation between two specified temperature thresholds (e.g.,
Tiedtke, 1993).    Then, condensation on ice can be calculated following the kinetic
approach, but the condensation on cloud drops still follows the saturation adjustment
in most models.    If the air is subsaturated but with the presence of cloud drops (or
cloud ice), the cloud drops (or cloud ice) are forced to evaporate to maintain the
equilibrium until they are all evaporated. As the saturation adjustment strategy
conventionally is not applied in subsaturated conditions for precipitation particles (e.g.,
raindrops, snow, etc.), it should be denoted as a partial equilibrium assumption.

The kinetic effect might have significant impacts on isotope fractionation and thus

there is a need to be considered in models.    For example, Hoffmann et al. (1998) tried
to consider the kinetic effect during deposition growth in the ECHAM AGCM model.
Due to the saturation adjustment assumption in ECHAM model, an effective factor,
which is function of temperature only, is used to express the kinetic effect (Jouzel and
Merlivat, 1984). In Wernet et al. (2011), the condensation on ice is also calculated with
an effective factor, but the condensation on cloud drops is in equilibrium fractionation.
In reality, deviation from equilibrium is rather common in cloud, and its magnitude
depends on factors such as updraft speed and hydrometeors' size spectra. These
factors usually are not considered in existing models but are included in the NTU
scheme.

Key parameters such as the HDO saturation vapor pressure, $P_{s,HDO}$, and diffusion

coefficient, $D_{HDO}$ are modified to handle HDO in the NTU scheme.    The HDO
saturation pressure, which is needed for the kinetic mass transfer calculation in Eq. (5),
can be obtained by equating Eq. (3) to Eq. (4). The derived HDO saturation vapor
pressure is generally lower than that of $H_2O$, and the differences increase as
temperature gets lower (Fig. 2). The mass diffusivity of HDO in air, $D_{HDO}$, in Eq. (5)
was obtained based on the relationship proposed by Hirschfelder et al. (1954):

$$D_x \propto \frac{m_{Air}+m_x}{m_{Air}m_x} \tag{7}$$

where $x$ represents any gas molecule.    Assuming that the proportionality
constants are the same for $D_{HDO}$ and $D_{H_2O}$, one can obtained the following:

$$\frac{D_{HDO}}{D_{H_2O}} = \frac{\frac{m_{Air}+m_{HDO}}{m_{Air}m_{HDO}}}{\frac{m_{Air}+m_{H_2O}}{m_{Air}m_{H_2O}}} \cong 0.9676 \tag{8}$$

with which we can relate $D_{HDO}$ to $D_{H_2O}$.

In Eq. (6), the activity of water stable isotope depends on the composition of the

particle.    For ice particles, the model cannot trace the history of water stable isotope
deposition and thus cannot distinguish between the surface layer from the inner core
of the ice particles.    Therefore, the water stable isotope activity of ice-phase
hydrometeor is assumed to depend on its bulk composition (i.e., assuming well-mixed).
In reality, however, there is no homogenization of isotopes in ice particles due to the
low diffusivities of molecules in ice.    Blossey et al. (2010), Pfahl et al. (2012) and
Dütsch et al. (2016) dealt with this problem by setting the ice particle's isotope ratio
equal to that produced by vapor deposition. This is an effective approach as only the
most recently deposited ice is exposed to the vapor. However, during evaporation the
mass exchange depends heavily on the residual composition, making the treatment
rather tricky. Before a better solution is devised, this study adopted the bulk
composition approach for both condensation and evaporation processes.
2.2    Simulation Setup

Frontal systems are not only rich in cloud microphysical processes but also involve

air-mass transitions and atmospheric circulation. As a result, they are ideal for
evaluating the relative contribution of various physical processes to isotopic
fractionation. The case selected for this study is a frontal system that passed through
northern Taiwan on 11 June 2012, with moderate to heavy rainfall from the night of
11 June until noon on 12 June. Special focus will be placed on northern Taiwan because
of the availability of isotope measurements for verification.
The simulation domain is shown in Fig. 3. The resolution of the coarse domain
was set at 81 km, covering the region from 90° to 150°E and 0° to 50°N. The resolutions
of the nested domains were set at 27 km, 9 km, and 3 km. The innermost domain
covers Taiwan and the surrounding ocean. Twenty-eight vertical layers were used,
eight of which were below 1.5 km (roughly the height of the planetary boundary layer),
with a maximum model height at 50 hPa. For the initial and boundary conditions, we
applied the National Centers for Environmental Prediction (NCEP) Final Global analysis
(FNL) data with a 1° by 1° resolution. FNL data for wind properties and temperatures
were nudged into domains 1 and 2 only every 6 h for better simulation of the
meteorology.    The physical options used in the WRF model included the NTU
microphysical scheme, the rapid radiative transfer model (RRTM) longwave and
shortwave radiation scheme (Mlawer et al., 1997), and the Yonsei University (YSU)
planetary boundary layer scheme (Hong et al., 2006). Cumulus parameterization was
turned off in the simulations.
To examine different factors that control the water stable isotopes concentration,
six simulations were conducted: the control run (CTRL) used the kinetic approach for
cloud microphysical processes; the EQ run used the thermal equilibrium approach;
NoIce was conducted to examine the differences between liquid- and ice-phase
fractionations; NoLnd inspects land-sea contrast of water vapor sources; and NoVh is
for investigating the vertical exchange of isotope composition between lower and
upper troposphere. We also conducted a blank test (NoFrac) in which isotopic
microphysical fractionation was turned off. Descriptions of these numerical
experiments is listed in Table 1.
The isotopic value for water vapor or condensates is conventionally expressed as
δD (conventionally expressed in ‰):
$$\delta D = \left( \frac{R}{R_{SMOW}} - 1 \right) \tag{9}$$
where $R$ is the $\frac{HDO}{H_2O}$ ratio in the sample, and $R_{SMOW}$ is the Vienna Standard Mean Ocean
Water isotopic ratio (Craig, 1961).   The lower boundary condition of $\delta D$ over land
and ocean are calculated by relating HDO flux to $H_2O$ flux according to Eqs. (3) and (4).
In such a conversion, the ratio $R_l$ over land is set to be that in surface precipitation
according to observed mean climatology in June from the Global Network of Isotopes
in Precipitation (GNIP) (Johnson and Ingram, 2004; Rozanski et al., 1993). The obtained
initial near-surface distribution of water vapor δD ($\delta D_V$) is shown in Fig. 4a.
The vertical distribution of initial atmospheric water stable isotope
concentrations (Fig. 4b) was obtained from the NASA TES-Aura level-3 data
(http://tes.jpl.nasa.gov/data/products/).   We took the data for the month of June
and averaged over years 2006-2012.   Although the concentrations of water vapor (QV)
and HDO (QIV) usually decrease exponentially with height, their ratios (i.e., QV:QIV)
vary rather linearly with height.   So, for areas over land, the vertical profile is fitted
as the following:
$$QIV(z) = \left( \frac{QIV_{srf}}{QV_{srf}} \right) \cdot (-4.940699 \cdot 10^{-5} \cdot z + 1.128299) \cdot QV(z) \tag{10a}$$
where $QIV_{srf}$ and $QV_{srf}$ are near surface value of QIV and QV, respectively.   For
marine environments, the profile is fitted as:
$$QIV(z) = \left(\frac{QIV_{srf}}{QV_{srf}}\right) \cdot (-5.005261 \cdot 10^{-5} \cdot z + 1.134024) \cdot QV(z) \qquad (10b)$$

Note that these formulas apply only to the free troposphere; within the planetary
boundary layers, QIV is assumed to be well mixed (see Fig. 4b for the full profiles).

2.3    Observations

The isotopic water vapor and rainwater δD data from 11–12 June, 2012, were

recorded using a cavity ring-down spectroscopy analyzer (CARDS, Picarro L2120-i),
following Gupta et al. (2009). The measurement of rainwater was conducted on the
fourth floor of the building of the Department of Geography, National Taiwan
University (NTU, 25.02°N, 121.53°E). The isotopic water vapor measurements were
conducted at Academia Sinica (AS), which is about 10 km east of the rainwater
collection site. The two sites are marked as N and A, respectively, in Fig. 5a. The
uncertainties in δD for liquid and vapor samples were found to be less than 0.3‰ and
1.0‰, respectively (Laskar et al., 2014). The precision of water vapor concentration
measurements made using a Picarro CRDS is less than 100 ppmv (Crosson, 2008); this
is applicable to all of the data presented here. In addition to these experimental data,
the NCEP Reanalysis II (R2) data and precipitation data from the Central Weather
Bureau of Taiwan (https://www.cwb.gov.tw/eng/index.htm) were used to verify the
simulations. Unfortunately, the NASA TES-Aura satellite daily data during this case not
available for verification over the studied region.

3    Results
3.1    Model verification

Comparison of the model results with the NCEP R2 data shows that the model

captured the locations of the cold front and associated low-pressure system
reasonably well; the front was over the East China Sea on 11 June and moved to Taiwan
on 12 June (Fig. 6). However, the simulated precipitation was generally lower than
observed, especially over northwestern Taiwan (Fig. 5a). Additionally, the first peak in
rainfall during the early morning of 11 June (Fig. 5b), was not obvious in the simulated
results. The impact of these discrepancies will be discussed in section 4.
The observed $\delta D_V$ was about $-90\sim-120‰$ during the pre- and post-frontal
periods, and decreased to a minimum of $-160‰$ on 12 June. The simulated $\delta D_V$
($-70\sim-100‰$) were about 20‰ higher than observed during the pre- and post-frontal
periods (Fig. 7), whereas the minimum $\delta D_V$ of $-150‰$ was slightly higher than the
observed during the rainy period. Observation of $\delta D$ in precipitation ($\delta D_L$) was
available only after 09:00 (local time) on 12 June (Fig. 7b). It decreased slightly from -
70 to $-90‰$ before 16:00 and then recovered to around $-30‰$ by the evening of 12
June. The simulated minimum is also around $-90‰$, but occurred a few hours earlier
than observed. The classic amount effect cannot be assessed from
observations. For model simulations, the simulated $\delta D$ in precipitation (Fig. 7b)
decreased with precipitation occurred (Fig. 5b). The negative correlation is similar to
the amount effect in other studies. Overall, the model captured reasonably well the
pattern and magnitude of changes in $\delta D$ during the frontal passage, except that the
timing is off by a few hours.

3.2 Factors affecting isotopic fractionation
The simulated spatial distribution of $\delta D_V$ in Fig. 8a and 8d show two main zones
of minimum $\delta D_V$, one over mid-latitudes and the other over the latitudes of Taiwan.
The former is mainly due to low $\delta D$ of surface vapor source (cf. Fig. 4a); whereas the
latter is associated with the frontal rainband, and corresponds to the observed minima
shown in Fig. 7a. At a first glance, one may deduce two main causes for the minima.
Firstly, the near-surface air in the frontal zone is basically of continental origin, where
the $\delta D_V$ is lower than over the oceans (cf. Fig. 4a). Secondly, precipitation microphysics
inside the frontal system caused a strong reduction (fractionation) in $\delta D$ of
hydrometeors as can be seen in Figs. 8e and 8f; therefore, the evaporation of
hydrometeors would produce low $\delta D_V$ in the lower troposphere.   The above results
are in agreement with the finding of Dütsch et al. (2016), who pointed out that
horizontal transport determines the large-scale pattern of water stable isotope in both
vapor and precipitation, while fractionation and vertical transport are more important
on a smaller scale, near the fronts. Note that the location of the hydrometeor's $\delta D$
minima at 500hPa and 850hPa is shifted due to the structure of the frontal system.
However, the relatively high $\delta D_V$ behind (to the north of) the frontal system may seem
a bit strange, as the air mass there should be of continental origin.   This suggests
more complicated mechanisms.   Besides the water vapor source and microphysical
fractionation, other factors such as the initial vertical distribution may also contribute
to the variation in $\delta D$ values. So, in order to decipher all possible controlling factors
and to evaluate their relative contributions, we need to examine results from the 5
sensitivity experiments that listed in Table 1.

The most obvious differences between the CTRL and other simulations in terms

of $\delta D$ in the vapor ($\delta D_V$) and liquid ($\delta D_L$) phases at 850 hPa occurred near the front
because that is the location of the richest microphysical fractionation and largest
contrast in air mass properties (Fig. 9). Isotopic fractionation due to phase change in
the CTRL run was weaker than that calculated in the EQ run (Fig. 9a), because the
isotopic compositions were not always in equilibrium between the different phases in
the CTRL run. That led to slower isotopic fractionation under severe phase changes.

The vertical distribution of $\delta D_V$ between the CTRL and EQ runs over northern

Taiwan (121-123°E, 25-27°N) is shown in Fig. 10a. The differences in water vapor $\delta D$ at
around 850 hPa or higher prior to the passing of the front (point A, Fig. 10a) are
associated with cloud formation due to mesoscale lifting in the warm air sector. When
the frontal system passed through northern Taiwan in the early morning of 12 June,
low $\delta D_L$ extended almost down to the surface.   The $\delta D_L$ in the EQ run was about 30‰
lower than that in the CTRL run during this period. These results suggest that the
equilibrium assumption may lead to large biases in $\delta D$ for a synoptic-scale weather
system as mentioned in other studies (e.g., Risi et al., 2010), and kinetic calculation is
crucial to isotope modeling.

The degree of isotopic fractionation is related to temperature. As the ratio

between the saturation pressure of $H_2O$ and HDO in different phases deviate more
from unity at lower temperatures (cf. Fig. 2), higher degree of fractionation will occur
at lower temperatures. The significance of ice-phase fractionation is tested with the
NoIce run, for which the saturation vapor pressure of ice-phase HDO was assumed to
be the same as that of the liquid phase, which leads to weaker HDO vapor deposition
on ice. The resulting differences in $\delta D_V$ are small near the surface (Figs 9b and 10b) but
become significant at higher altitudes where the ice fractionation deviate more from
that of liquid.   Reduced $\delta D$ in the ice phase ($\delta D_I$) can be seen immediately above the
0°C level (Fig. 10b), causing more heavy water isotopes to remain in the gas phase and
then transport to higher altitudes. This results in an elevated $\delta D$ in both the vapor and
the ice phase. The increase in $\delta D_V$ and $\delta D_I$ can reach over 50‰ and 30‰, respectively,
near the tropopause. Such changes may also affect the lower troposphere, because
snow and graupel particles may fall to lower levels and bring down high $\delta D_I$ water.
The amount of changes due to such gravitational sorting depends on whether
snow/graupel were formed in the lower or higher mixed-phase zone; the former leads
to lower $\delta D_I$, while the latter increases it. However, the changes were generally within
10‰. Due to the temperature dependence of the isotopic value and the structure of
the atmosphere, ignoring the difference between liquid and ice-phase fractionations
will lead a vertical redistribution of the isotopes.
The initial and boundary conditions are also important in determining the isotope
levels. Based on the IAEA data, precipitation $\delta D$ decreases from marine to inland areas,
indicating that the water source is important in determining the initial water stable
isotope content. In the NoLnd run, the initial $\delta D$ over land was set to be the same as
that over the ocean, and this resulted in a higher $\delta D$ not only over land but also in the
frontal system (Fig. 9c). Ahead of the front, the vapor-phase $\delta D$ in the NoLnd run
increased by about 40‰ relative to the CTRL run. The initial vertical distribution of $\delta D_V$,
which was based on satellite data, showed large vertical decay into the free
troposphere. In the NoVh run, the initial $\delta D$ in the free troposphere is assumed to be
the same as that in the planetary boundary layer. This caused 20-50‰ overestimation
of $\delta D_V$ at near surface (Fig. 9d).
When the observed and simulated $\delta D_V$ at AS and precipitation $\delta D_L$ at NTU are
compared (Fig. 11), one can see that the full simulation (i.e., the CTRL run, red line) is
rather close to the observation in terms of the peak values during the time of frontal
passage (06:00-12:00 LST on June 12). In contrast, the decrease in $\delta DV$ was
overestimated by 11‰ in the equilibrium run and underestimated by 28 and 34‰ in
the NoLnd, and NoVh runs, respectively. The simulated $\delta D_V$ in the NoIce run is rather
close to that in the CTRL run, which is consistent with the vertical profile shown in Fig.
10b, suggesting that the ice-phase process does not have a significant effect on $\delta D$ at
lower altitudes; however, the changes in the upper troposphere are significant. The
importance of microphysical fractionation is elucidated with the NoFrac run (grey line
in Fig.11), which yields 25‰ and more than 50‰ differences in the minimum $\delta D_V$; and
$\delta D_I$, respectively.

4    Discussion

Combining the observations and simulations results    of the water stable

isotopes can be used to understand the water cycle. From the observed $\delta D_V$ decreased
after 06:00 on 12 June (black line in Fig. 11), much later than the onset of the
precipitation. The observations suggested that the source of water vapor before this
time is the ocean (Wang et al., 2016) and that the microphysical processes related to
the precipitation did not substantially affect $\delta D_V$ during this period.    Model
simulations can help with further understanding of such isotopic fractionation. The
$\delta D_V$ on 11 June varied little among different tests (Fig, 11) because the airmass was
from nearby areas (i.e., no significant advection effect) and no cloud microphysical
processes occurred during this period.    However, water vapor $\delta D_V$ decreased from -
80 to -100‰ at midnight of 11-12 June in the control run, but not in the NoLnd run,
indicating that the decreases in $\delta D_V$ was due to advection of the continental airmass.
When the front passed through during the early morning of 12 June, $\delta D_V$ decreased
from −100 to −170‰ in the control run but not in the NoFrac run (grey line in Fig.10),
indicating that the additional differences was caused by cloud microphysical processes.
After the passage of frontal system, $\delta D_V$ returned to its background level, around
−80‰. The results of these sensitivity tests suggest that the changes in $\delta D_V$ due to
cloud microphysical processes, initial vertical distribution, and lower boundary
conditions are of a similar order, and are all important to isotopic fractionation.

Although the model seems to adequately reproduced changes in $\delta D$ in this frontal

case, there are some minor inconsistencies between the simulation results and
observations. Some discrepancies originated from the meteorological model itself and
the initial meteorological conditions, which caused inaccuracies in the intensity or
timing of surface precipitation. In fact, most models including ours failed to simulate
the strong precipitation over land for this system (Wang et al. 2016). The observed
water vapor and precipitation δD values were not in phase, and the water vapor δD
decreased prior to precipitation (black line in Fig. 11). In contrast, the decreases in the
simulated precipitation and water vapor δD were almost simultaneous, starting
around 03:00 on 12 June (red line in Fig. 11). This again suggests that the model missed
an earlier local convection system occurred during the early morning, such that the
simulation can reflect only the δD variation due to the frontal system. The simulated
$\delta D_V$ decreased and returned to its previous level earlier than the observed $\delta D_V$
(~03:00-10:00 compared to ~07:00 -13:00). This also suggests that the arrival time of
the frontal system to Taipei was earlier than observed, although the speed of the
simulated system was close to that of the observed one, taking about seven hours to
pass through Taipei.
Uncertainties may also exist in the observation data, as the vapor and
precipitation measurements were taken at different locations, separated by about 10
km.    A comparison of precipitation at different sites (Fig. 12; NanGang station is close
to AS, and GongGuan station is close to NTU) suggests that the difference in sampling
locations would not significantly affect the results in this study. Another uncertainty is
the parameterization of isotopic fractionation factor $\alpha$. In this study, the temperature
dependence of $\alpha_{l-v}$ and $\alpha_{s-v}$ were adopted from Horita and Wesolowski (1994)
and Ellehoj et al. (2013) , respectively.    In most models, the formulation for ice/vapor
by Merlivat and Nief (1967) is still used. From Fig. 2 one can estimated that the
differences in $\alpha_{s-v}$ between Ellehoj et al. (2013) and Merlivat and Nief (1967) are
around 1% between -10~-20$^\circ$C and 4% at -40$^\circ$C; whereas the differences of
$\alpha_{l-v}$ between Horita and Wesolowski (1994) and Merlivat and Nief (1967) are less
than 1%.
There are also uncertainties in the treatment of microphysical processes. The
isotopic value for water vapor at the lower boundary condition was assumed to be in
equilibrium with surface precipitation in this study. Rangarajan et al. (2017) analyzed
the isotopic ratios in water vapor from measurements over Taipei, and they found that
isotopic values were not always in equilibrium. This suggests that the assumed lower
boundary condition might not always be applicable for Taipei. Moreover, since the
lower boundary condition can be affected by fresh precipitation, $\delta D_V$ might decrease
after the precipitation event which brings in low $\delta D_L$ to the soil; yet, our model does
not update the surface $\delta D_V$ flux accordingly.    This might partially explain the
discrepancy in $\delta D_V$ after the frontal passage that shown in Fig. 7a. In addition, the
evaporation from the ocean is assumed as in equilibrium between liquid and vapor
phases. This assumption may also affect the simulation of $\delta D$ in the model, and the
process needs to be explicitly considered in the future study. Finally, whether the
nonequilibrium effects are important for the second-order isotope parameter,
deuterium excess, is an interesting subject worthy of further investigation by including
the description of $\delta^{18}O$ isotope in the model.

5    Conclusion
Exploring physical processes controlling the stable isotopic composition of water,
including details such as water vapor source, atmospheric circulation, and cloud
microphysical processes, is useful for understanding the water cycle. In this study, we
modified the NCAR WRF model to understand the role of different factors in the
fractionation of the stable isotopes of water. The experimental stable isotope thermal
equilibrium data were converted into isotope saturation vapor pressure, which was
then used in the two-stream Maxwellian kinetic equation for calculating the
condensation/evaporation or deposition/sublimation of HDO, in parallel with that for
$H_2O$. Mass conservation was also considered explicitly for the collection processes as

472 well as during freezing/melting.

473   A frontal system event was selected to reveal the complexity of isotope

474 fractionation. The model captured the location of the front adequately, although the

475 estimated precipitation was less than observed. The simulated results showed fairly

476 good agreement with water vapor and rainwater stable isotope measurements, and

477 suggested that the decreases in water vapor δD before the front arrived in Taiwan was

478 due to an airmass of continental origin. When the front passed during the early

479 morning of 12 June, both the water vapor sources and the cloud microphysical

480 processes contributed to a decrease in water vapor δD, which returned to a

481 background levels after the front had passed.

482   Additional sensitivity experiments showed that the thermal equilibrium

483 assumption commonly used in earlier studies might significantly overestimate the

484 decrease of mean δD by about 11‰, while the maximum difference can be more than

485 20‰, during the precipitation event. Cloud microphysical processes, including ice-

486 phase processes, have substantial effects on isotopic fractionation, especially on the

487 vertical redistribution of isotopes. Furthermore, the sensitivity tests suggest that the

488 initial vertical profile and the land–sea contrast in surface sources are quite important

489 in simulating atmospheric stable isotopic composition, and should be estimated from

490 observations such as satellite data, without which the underestimation in the decrease

491 of water vapor δD could reach about 34 and 28‰, respectively. The problem in

492 determining the activity of water stable isotope in ice particles without knowing the

493 inhomogeneity of chemical composition in the bulk ice, as mentioned at the end of

494 section 2.1 is another issue worthy of further study. To accommodate the different

495 conditions between condensation and evaporation, it might be feasible to assume that

496 the water stable isotope activity is determined by the vapor phase during

497 condensation following the approach of Blossey et al. (2010), Pfahl et al. (2012) and

Dütsch et al. (2016); whereas for the evaporation process, one may assume a well-
mixed bulk composition for determining the isotope activity as done in this study. In
summary, this study suggests that a better understanding in the relationship between
water stable isotope variation and hydrological cycle can be achieved with a
combination of multi-platform observations and detailed cloud model simulations.

Acknowledgment. This study was supported by projects MOST 105-2119-M-002 -028
-MY3, 105-2119-M-002-035, 106-2111-M-001-008, and 107-2111-M-001-006. The
suggestions provided by anonymous reviewers are highly appreciated.

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

Table 1. Description of the six numerical experiments conducted in this study.

| Abbreviation | Description |
|---|---|
| CTRL | All processes are included. |
| EQ | The isotopic fractionation between different phases is in thermodynamic equilibrium. |
| NoIce | The isotopic fractionation between solid and other phases is the same as that of liquid and other phases (i.e., assuming the vapor pressure of solid HDO is the same as that of liquid HDO). |
| NoLnd | The initial $\delta D$ over land is set to be the same as those over the ocean. |
| NoVh | The initial $\delta D$ in the free troposphere was equal to that in the planetary boundary layer (i.e., no vertical gradient). |
| NoFrac | No isotopic fractionation considered in cloud microphysical processes (i.e., HDO is treated as a tracer). |



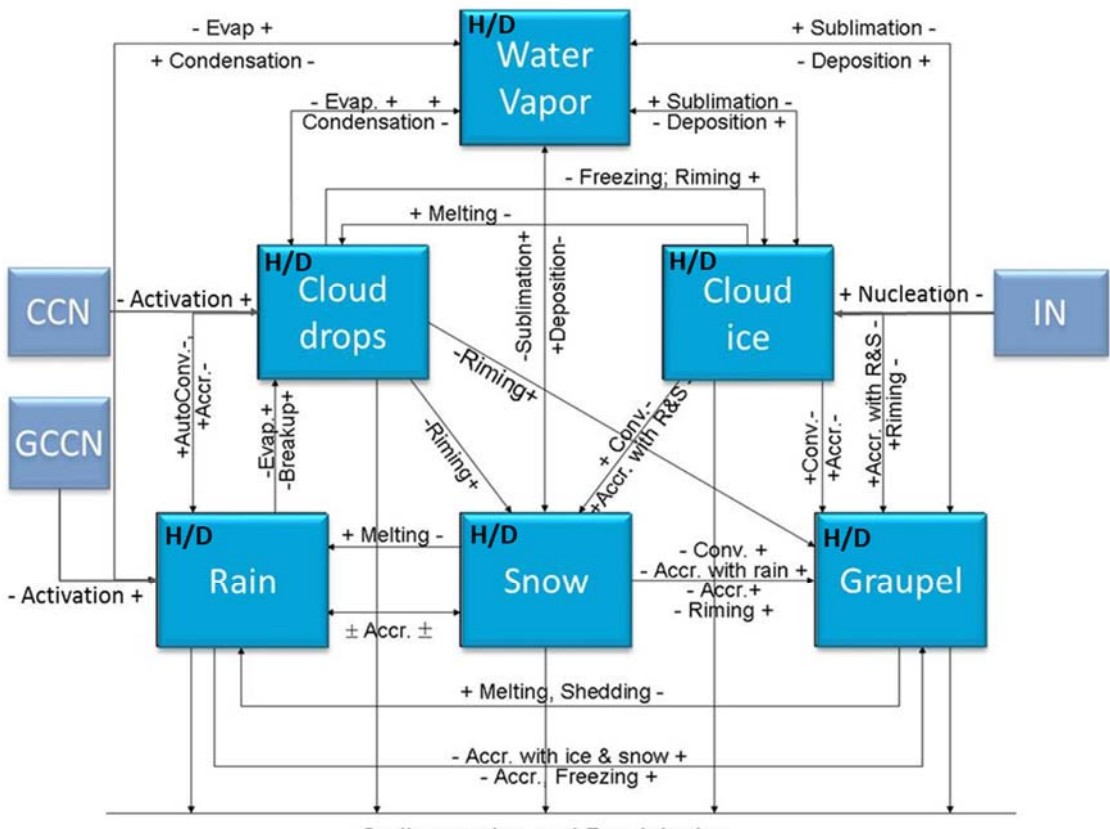


Figure 1. Schematics of the modified NTU scheme. The blue boxes are the hydrometeor
categories considered in the model, and the H/D indicated that both $H_2O$ and HDO are
included. The arrows represent the microphysical conversion processes; and the light
blue boxes represent aerosol categories, including cloud condensation nuclei (CCN),
giant CCN (GCCN) and ice nuclei (IN). (Figure modified from *Cheng et al*. 2010)



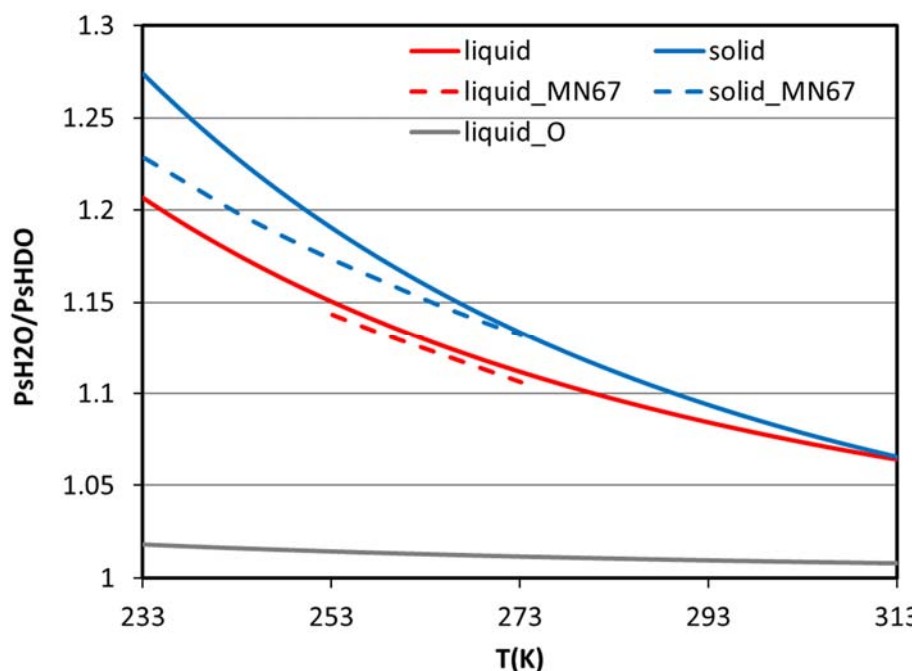


Figure 2. The ratio between saturation pressure of $H_2O$ and HDO in different phases (liquid: red line, solid: blue line) at different temperatures. The grey line is the ratio of [18]O based on Horita and Wesolowski (1994). The dash lines are the formulas from Merlivat and Nief (1967).




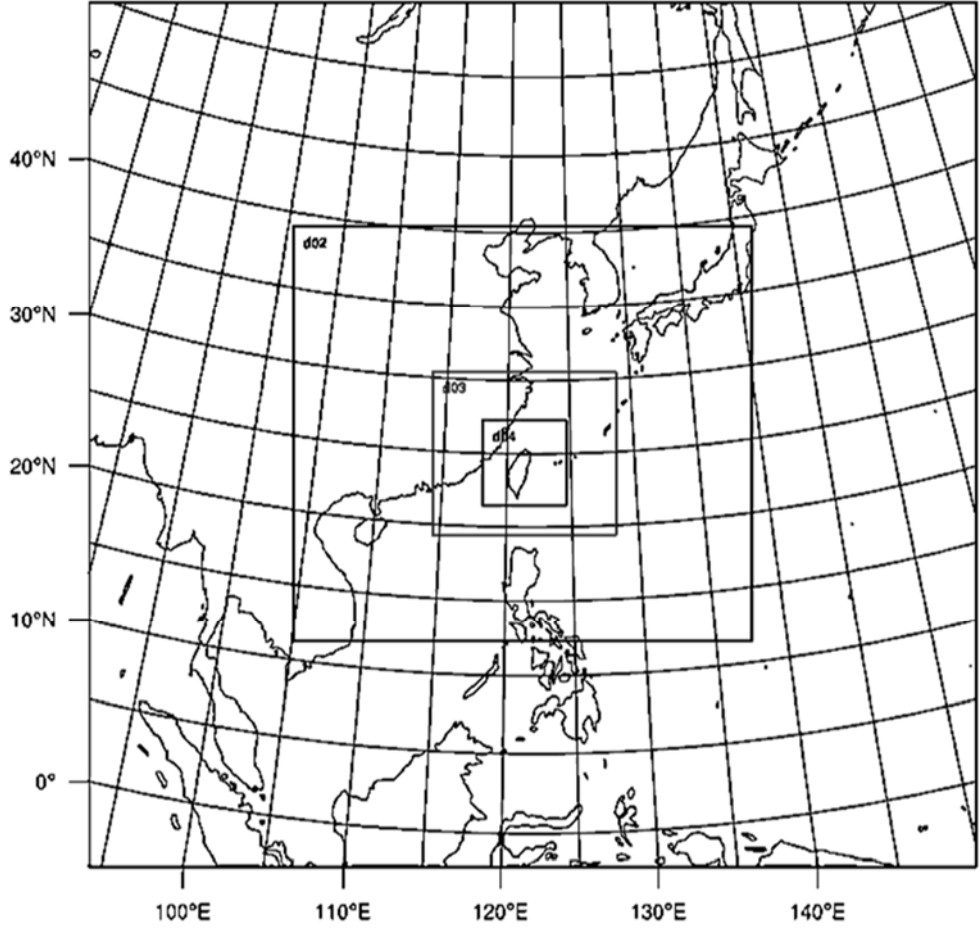


Figure 3. Map of the model domains for the simulations in this study. The resolutions
are 81, 27, 9 and 3 km in the outmost, 2nd, 3rd, and inmost domains, respectively.


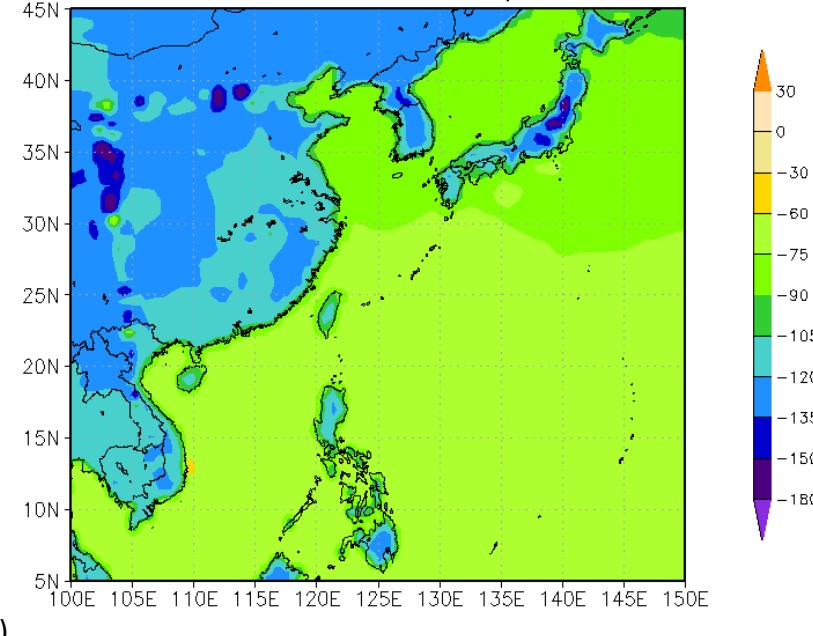

(a)

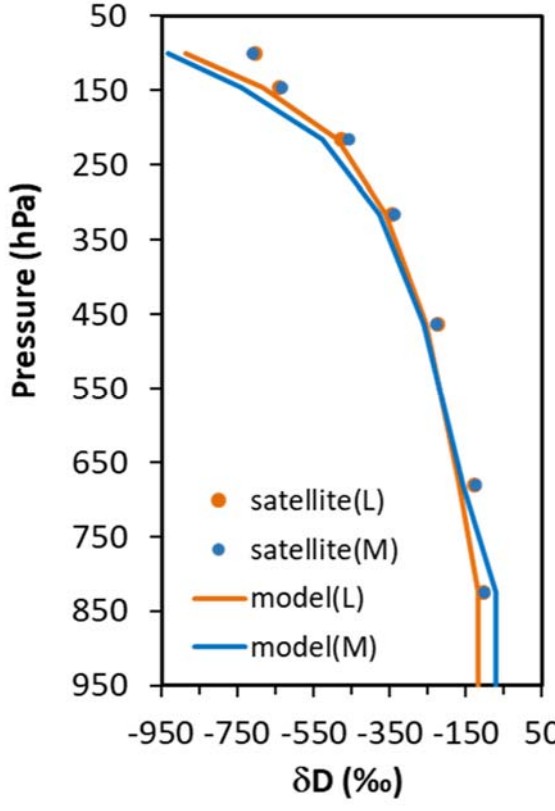

(b)

Figure 4. The initial distribution of water vapor δD (in ‰). (a) Surface distribution in
the coarse domain; (b) water vapor δD vertical profiles fitted from satellite data (dots).
Orange is for land (L), and blue is for marine (M)

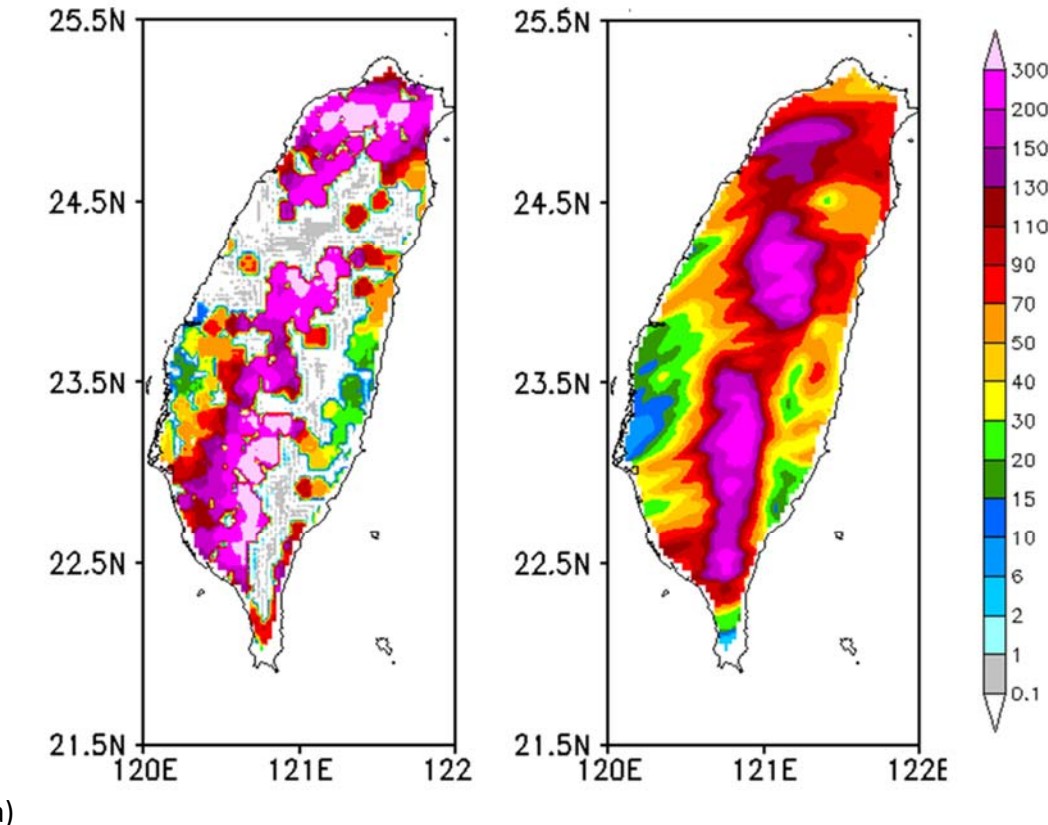

(a)

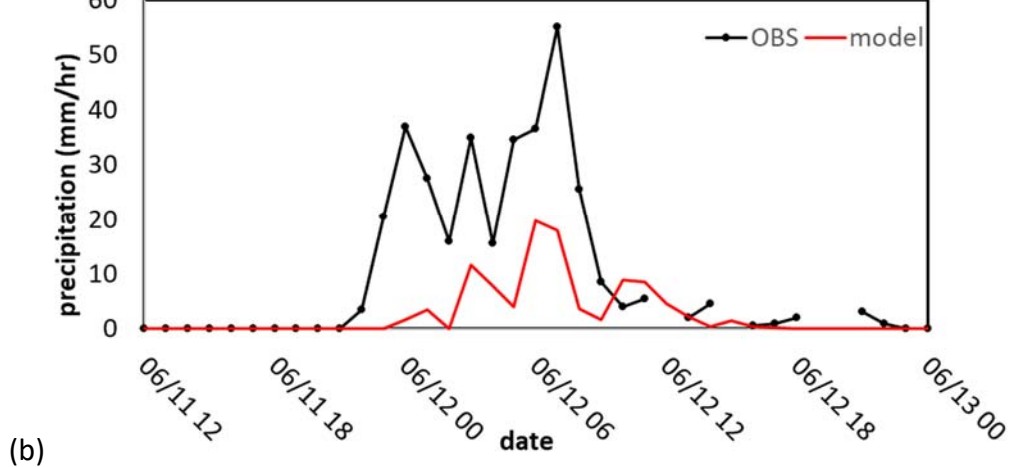

(b)

Figure 5. (a) Comparison between observed (left) and simulated (right) accumulated
precipitation (mm/hr) in Taiwan on 12 June 2012. Mark N and A denotes the location
of NTU and AS.    (b) Simulated (red line) and observed (black line) precipitation
(mm/hr) at Taipei station on 11-13 June 2012.


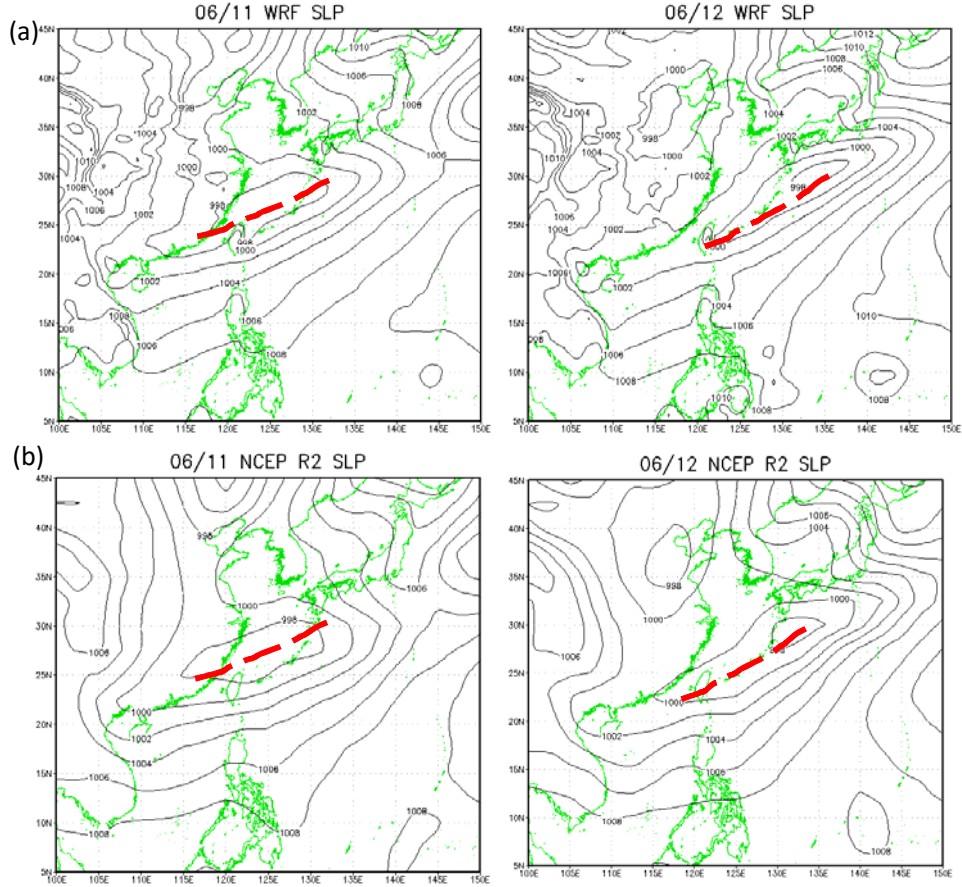


Figure 6. Comparison of (a) simulated sea level pressure (hPa) with (b) the NCEP
reanalysis data at 08:00 LST on 11 June (left) and 12 June (right), 2012. Frontal
position is indicated by the red dashed lines.

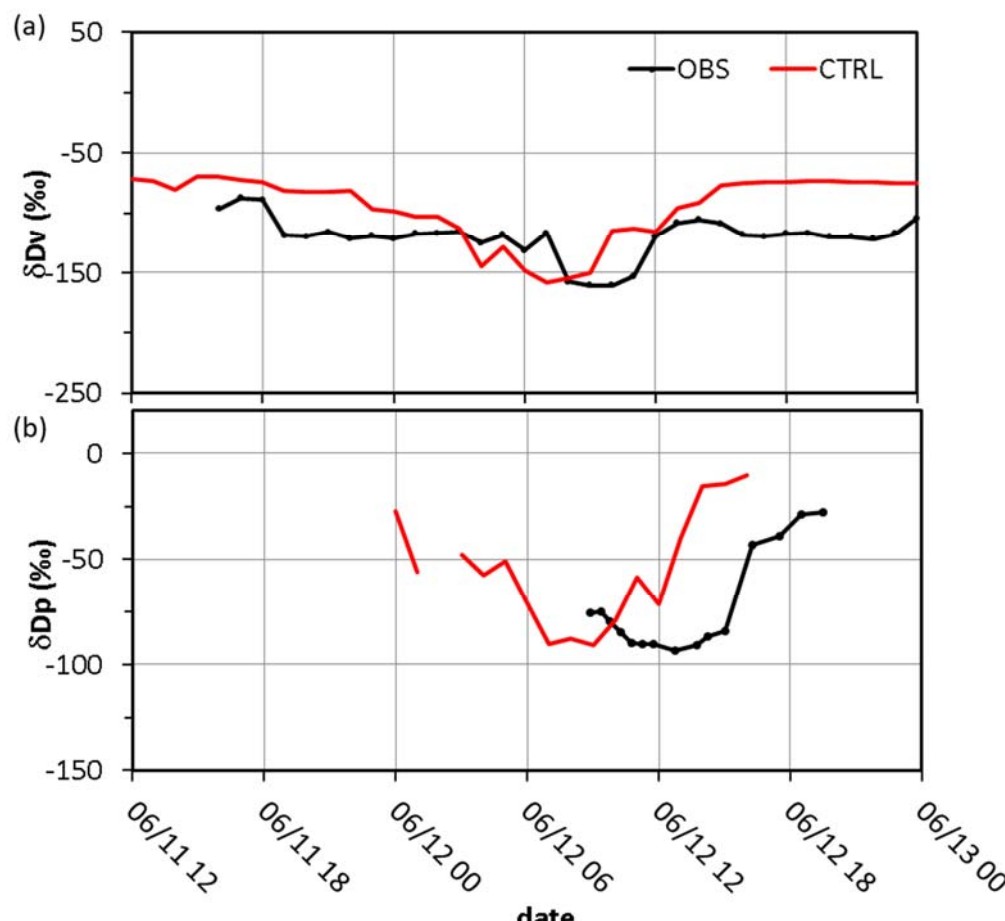



Figure 7. Simulated (CTRL: red line) and observed (OBS: black line) (a) water vapor
δDv (in ‰) at AS and (b) precipitation δDp (in ‰) at NTU on 11-13 June 2012.

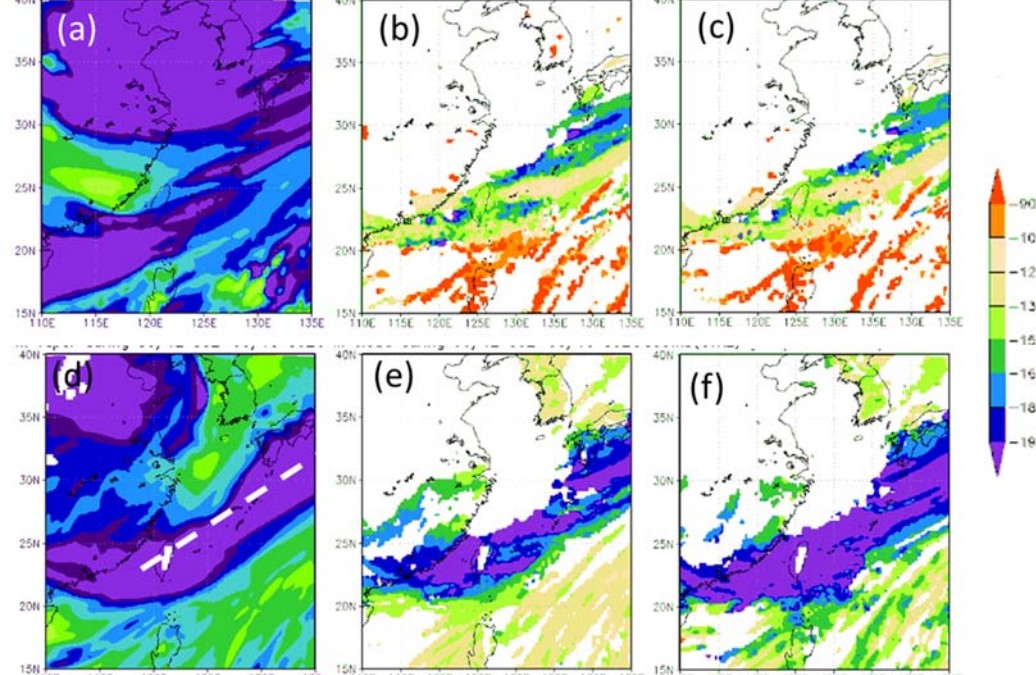


Figure 8. Simulated δD (in ‰) of water vapor (left), liquid-phase condensates
including cloud- and rainwater (middle), and ice-phase condensates, including cloud
ice, snow, and graupel (right) in the CTRL run at 500 hPa (a-c) and 850 hPa (d-f) on 12
June 2012.

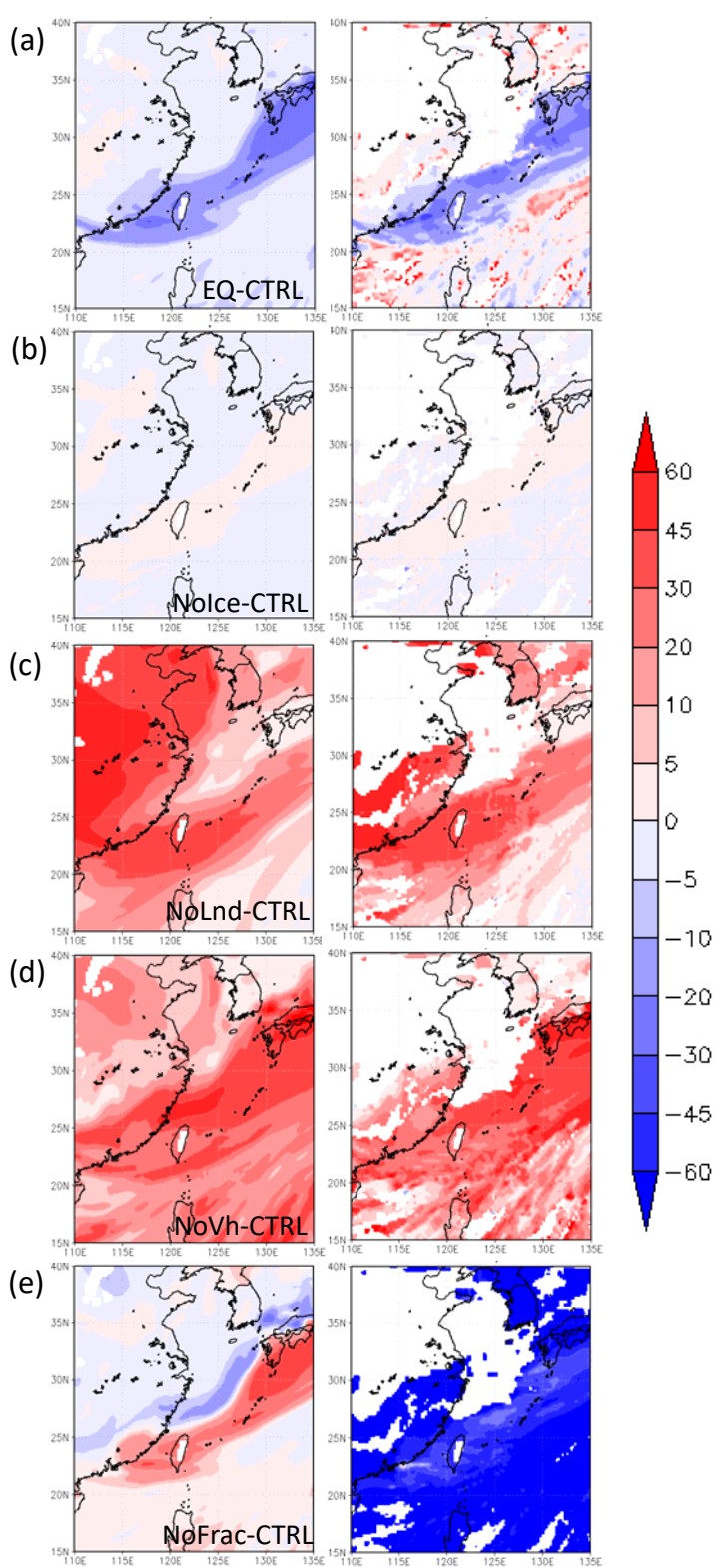

Figure 9. Difference in simulated δD (in ‰) of water vapor (upper) and liquid-phase
condensates, including cloud- and rainwater (lower), between CTRL and other runs: (a)
EQ-CTRL, (b) NoIce-CTRL, (c) NoLnd-CTRL, and (d) NoVh-CTRL, and (e) NoFrac-CTRL, at
850 hPa on 12 June 2012.

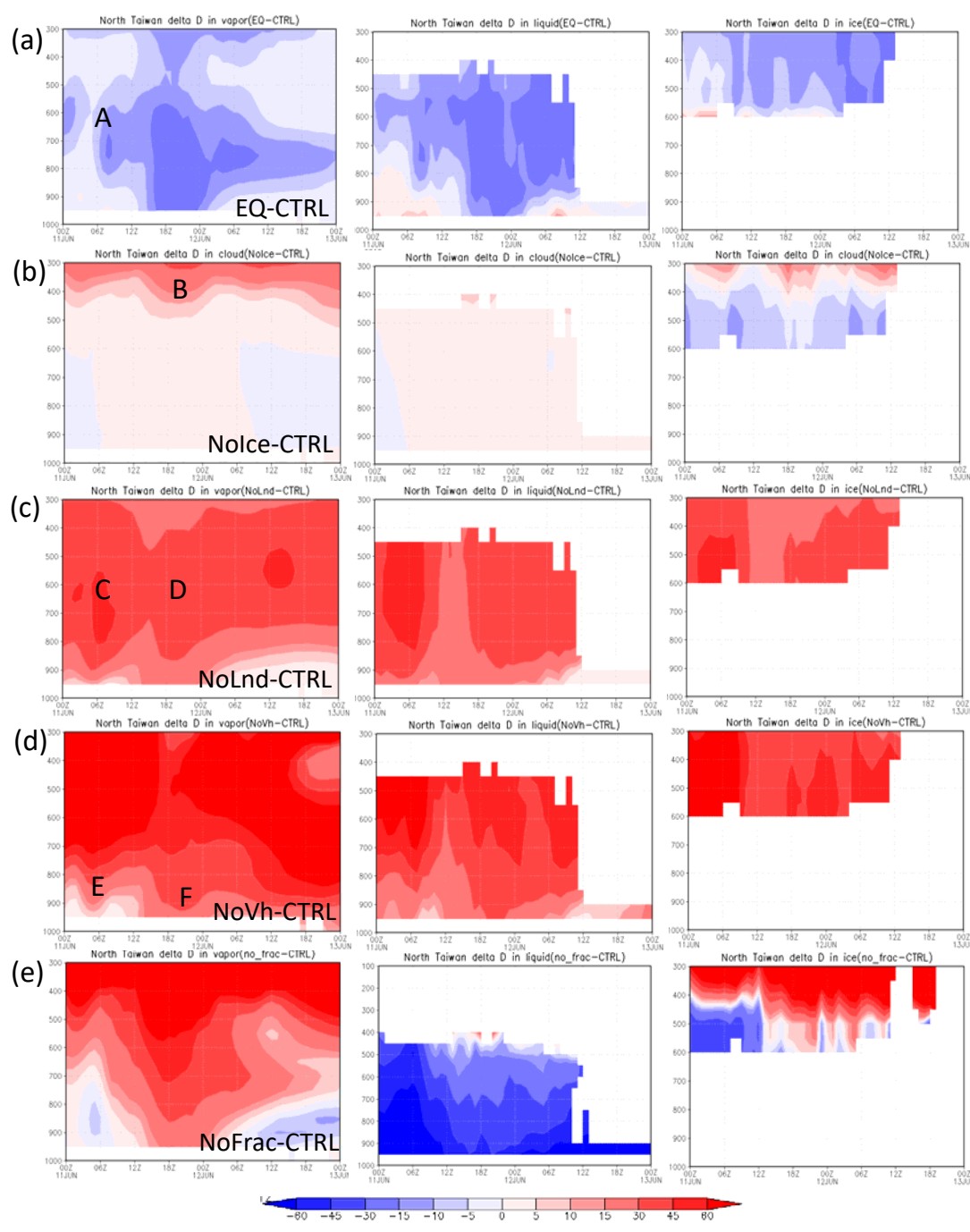

Figure 10. Time evolution of the vertical distribution of simulated water vapor δD (left), liquid-phase water (cloud water and rainwater; middle), and ice-phase water (including cloud ice, snow and graupel; right) over Northern Taiwan (121-123°E, 25-27 °N) in different simulations: (a) EQ-CTRL, (b) NoIce-CTRL, (c) NoLnd-CTRL, (d) NoVh-CTRL,and (e) NoFrac-CTRL on 11-12 June 2012.    The ordinate is pressure (hPa), and abscissa is time.

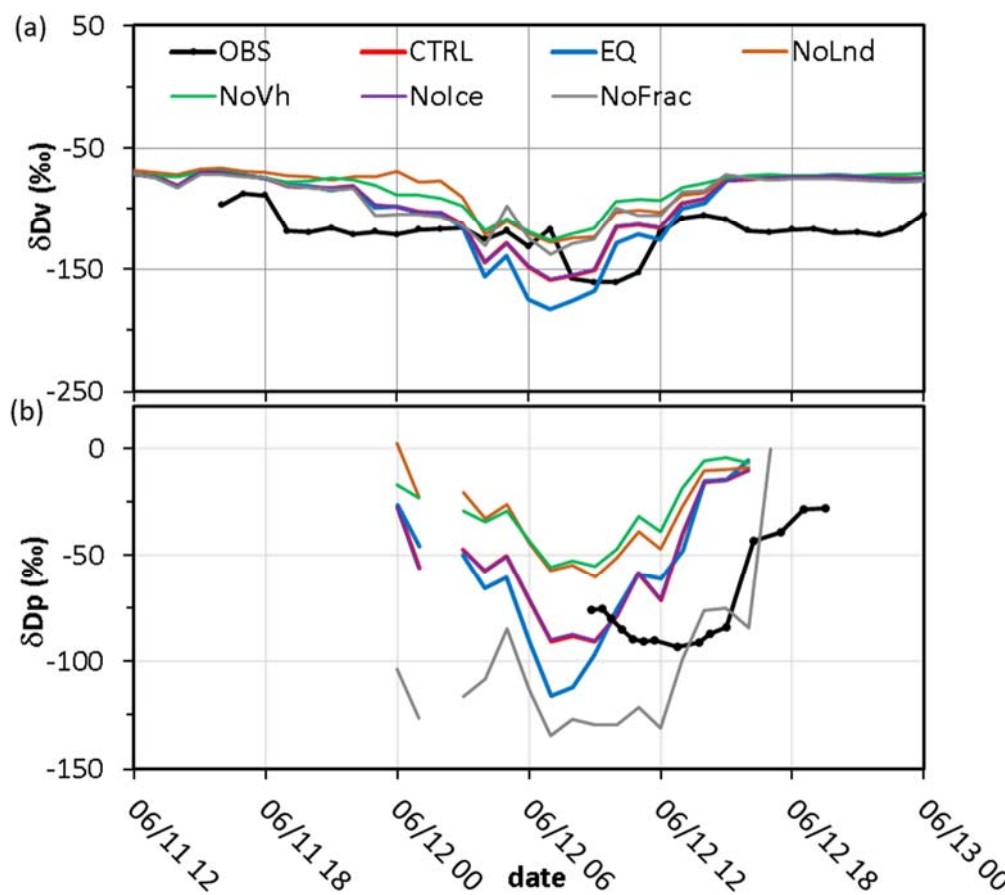


Figure 11. Same as Fig. 7 but for sensitivity simulations: the control run (CTRL, red line),
thermodynamic equilibrium run (EQ, blue line), no-ice run (NoIce, purple line), no-land
run (NoLnd, orange line), constant initial vertical profile run (NoVh, green line), and no
fractionation run (NoFrac, grey line).


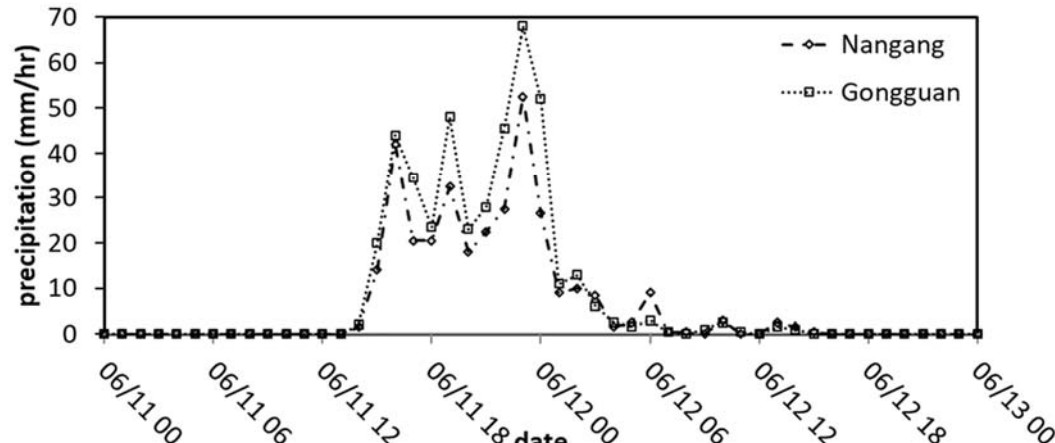

Figure 12. Precipitation (mm/hr) at NanGang (dash line) and GongGuan (dot line)
Stations on 11-13 June 2012.