# Peer review of "Kinetic mass-transfer calculation of water isotope fractionation due to cloud microphysics in a regional meteorological model"

_Atmospheric Chemistry and Physics, 2018_

## Referee Comment (RC1) · Anonymous Referee #1 · 17 Aug 2018

Summary Tsai et al. have made a case study of kinetic isotope effects in clouds during a precipitation event over Taiwan. The authors have implemented two isotopologues of water in the hydrological cycle of a regional model for this purpose. The study uses a growth model for the distribution of the droplet size in the cloud scheme. The conclusion is that kinetic effects have a significant effect on the isotope composition, and including these effects yields a result closer to observations.

Major Comments More studies of cloud process using high resolution models are most certainly needed to further our understanding, both of the cloud processes themselves, as well as for understanding the isotope fractionation processes. While this study at-

tempts to do these things I think that the manuscript in the current form only got the authors half way there.

1) I think the premises for this manuscript are not correct, or at least very imprecise. As I outline in the comments below (e.g. comment for L62-64) the authors use a 20 year old paper to motivate their study and generalize the research area. Quite some major studies have been published in the meantime. While the results by Tsai et al. might be correct and their model well functioning, the reader has little chance to know what is actually new in this study. Furthermore, it appears to me that the manuscript in it's present form might lead the reader to think that the approach of the study is more novel than it actually is (comments to L62-64 and L128). 2) Basic analysis of the relationship between precipitation and dD is missing. As I understand it, this isotope enabled version of the WRF model has not previously been published. For a new model I would expect more extensive validation. At least for the course-resolution model simulation. For the larger domain, much more data are available for a general evaluation. Then the detailed case study comes after, if the model shows reasonable performance. The authors mentions the amount effect once in the introduction never to return to it, neither in the analysis nor in the discussion. I suggest Kurita (2013) and Zwart et al. (2018) as a starting point. I also suggest a few more plots (see last two comments for Figures) that would be very helpful for the analysis and for the reader to have some fundamental understanding of the performance of the model. 3) When it comes to the writing, formulations are often not precise enough when describing specific processes (e.g. comments to L62-64, L86, L88, L128, L177-178, L259-260, L332-334). 4) The quality of plots and labels make it difficult for the reader to see the point the authors are trying to make. See comments for Figures.

Given these major comments and the specific comments below I cannot recommend this study for publication in its current form. I think it would be better to resubmit after recasting the study. I hope that the authors will take the time to do this, and maybe an updated study could be an interesting contribution to the topic of water isotopes in the

climate system.

Specific comments L18 Insert "obtain" after "to".

L22 The authors are careful to use the term "isotopocule", but the title says "water isotope". The colloquial, but technically incorrect, expression "water isotopes" is widely used. Whatever the authors choose, please be consistent in terminology. Generally, "isotopologue" is more widely used for water isotopes than "isotopocule", since they for water mean the same thing. A less heavy, but still correct, term would be "water stable isotopes".

L31 Replace "often" with "generally" and replace "variation" with "variations".

L32 Replace "location" with "space".

L36-43 It would be helpful to the reader to add specific references for specific processes discussed.

L62-64 Please be more precise when describing the what you mean by "partial or full equilibrium state". In Hoffmann et al. (1998) kinetic fractionation is taken into account during evaporation and snow formation, as well as partial evaporation of raindrops when the air is undersaturated. Kinetic isotope effects have been investigated in a number of GCM studies, here among: Schmidt et al. (2005), Risi et al. (2010), Werner et al. (2011) and Nusbaumer et al. (2017). Given the motivation and focus of the study, the authors owe it to the reader to highlight the existing literature and provide a precise description to elucidate what is novel in this study. Could it be that the authors are also not familiar with Yoshimura et al. (2010)? I would surely expect such a paper to be cited for this topic.

L72 Replace "Duterium" with "Deuterium"?

L72-74 Please reformulate. This means you only incorporate HDO and H216O? And if so, why? Kinetic effects have strong impact on the deuterium excess.

L86 With "saturation adjustment" do you refer to parameterizations of supersaturation? E.g. where supersaturation is a linear function of temperature below a threshold? Please provide an example and reference.

L88 "according to kinetic mass transfer principles" so this is not a parameterization, but explicit physics?

L98 I assume $H_2O$ means $H_2^{16}O$? Please specify.

L119-122 Have you tested the sensitivity to these formulations? In most models the formulation for ice/vapor by Merlivat and Nief (1967) is still used.

L128 As also noted by Robinson and Scott (1981) this equation is derived using similar arguments as Fick's law, and essentially says that the flux is proportional to the gradient of the concentration. The formulation of kinetic fractionation for bulk vapor is derived using Fick's law (e.g. Merlivat and Jouzel, 1984), and thus very similar to what is done in Eq. 5. My point being, that the real difference from your study to other isotope model studies is not that you take kinetic fractionation into account and they don't, because they actually do (see comment to L62-64). The difference is in the widely used bulkwater formulation, while you use a growth model for the distribution of the droplet size, and formulate the isotope fractionation accordingly (following existing principles). Giving the reader something to hold on to in comparing your work to previous work (e.g. Eq. 5 following similar principles as Fick's law, which other models base their kinetic isotope scheme on) is important to convey where your study is placed compared to other studies.

L145-147 Is this the motivation for the text L72-74?

L145 In Fig. 2, can you provide a comparison with liquid/solid phase vapor pressure using a the more common parameterizations found in GCMs?

L162 What about spin up of simulations? Longer spin up is usually required for isotope simulations.

L177-178 How is the nudging done? Spectral nudging or?

L197-215 So the observed d18O of sea water (e.g. LeGrande and Schmidt, 2006) is not used as lower boundary conditions, and your vapor isotope boundary conditions are derived from mean conditions? Sturm et al. (2005) showed that regional simulations are very sensitive to isotope boundary conditions, obtaining the best results with a nudged global model run as boundary conditions. Please provide some arguments how you can use this type of boundary conditions for a case study. For example, if the variability generally is small (please quantify) then it could be argued that, an observed mean "climatology" is representative. I am aware that you partly test the boundary conditions with the "NoLnd" run, but this is not what I'm asking for.

L234-236 That the model captures the location front is no surprise due to the nudging. Or is the nudging not constraining the model very tightly?

L259-260 You need to explain the continental effect somewhere, as you use this argument several times. This could for example be done in the introduction where the classic isotope effects are only touched upon in very general fashion.

L267 Replace "mechanimss" with "mechanisms".

L285-287 Please reformulate. Maybe simply replace "discrepancy" with "biases". Also, hasn't several studies already shown that this is generally true (e.g. Risi et al., 2010)? Does your study have smaller biases than other studies?

L288-291 This is one explanation, which sounds like an instantaneous process no matter the history of the air parcel. What about the progressive rain out of heavy isotopologues during adiabatic ascend?

L295 Replace "." with white space?

L297 Replace "heavier" with "heavy".

L332-334 Do you mean that there is a local marine source at the onset of the precipitation? In this context what does microphysical processes? I don't follow what the authors mean in this paragraph. Please rephrase.

L340 Replace "mid night" with "midnight".

Figures: * Please add axis labels and units to all graphs. E.g. Figure 4b has neither axis labels nor units, also no mention of units in the caption. * Labels and axis on Figure 9 and 10 are next to impossible to read. Please provide readable plots. * It is very difficult to see the differences between the curves (especially Figure 7 and 11) if you want to assess the discrepancy in time between the different curves. I suggest i) place subplots in upper and lower panels instead of side-by-side to "stretch" time ii) add axis grid in plot iii) make more readable axis labels that don't line up so close, for example by rotating them 45 degrees, and make it clear what is date and what is time. * It would be extremely helpful to plot dD and precipitation rates in the same plot to see if the timing of changes in dD and precipitation amount is similar in model and observations. * I suggest that the authors also add plots of dD vs precipitation and compare the model to observation and assess the classic amount effect. This is featured in many studies of isotopes in precipitation at low latitudes, and would give a way to compare to other studies.

References

Kurita, N. (2013), Water isotopic variability in response to mesoscale convective system over the tropical ocean, J. Geophys. Res. Atmos., 118, doi:10.1002/jgrd.50754.

LeGrande, A. N., and G. A. Schmidt (2006), Global gridded data set of the oxygen isotopic composition in seawater, Geophys. Res. Lett., 33, L12604, doi:10.1029/2006GL026011.

Nusbaumer, J., T. E. Wong, C. Bardeen, and D. Noone (2017), Evaluating hydrological processes in the Community Atmosphere Model Version 5 (CAM5) using stable isotope ratios of water, J. Adv. Model. Earth Syst., 9, 949–977, doi:10.1002/2016MS000839.

Risi, C., S. Bony, F. Vimeux, and J. Jouzel (2010), Water-stable isotopes in the LMDZ4 general circulation model: Model evaluation for present-day and past climates and applications to climatic interpretations of tropical isotopic records, J. Geophys. Res., 115, D12118, doi:10.1029/2009JD013255.

Robinson, N.F., Scott, W.T., 1981. Two-stream Maxwellian kinetic theory of cloud droplet growth by condensation. J. Atmos. Sci. 38, 1015–1026.

Schmidt, G. A., G. Hoffmann, D. T. Shindell, and Y. Hu (2005), Modeling atmospheric stable water isotopes and the potential for constraining cloud processes and stratosphere-troposphere water exchange, J. Geophys. Res., 110, D21314, doi:10.1029/2005JD005790.

Sturm, K., G. Hoffmann, B. Langmann, and W. Stichler (2005), Simulation of delta O-18 in precipitation by the regional circulation model REMOiso, Hydrol. Process., 19(17), 3425–3444, doi:10.1002/hyp.5979.

Werner, M., P. M. Langebroek, T. Carlsen, M. Herold, and G. Lohmann (2011), Stable water isotopes in the ECHAM5 general circulation model: Toward high-resolution isotope modeling on a global scale, J. Geophys. Res., 116, D15109, doi:10.1029/2011JD015681.

Yoshimura, K., M. Kanamitsu, and M. Dettinger (2010), Regional downscaling for stable water isotopes: A case study of an atmospheric river event, J. Geophys. Res., 115, D18114, doi:10.1029/2010JD014032.

Zwart C, Munksgaard NC, Protat A, Kurita N, Lambrinidis D, Bird MI. The isotopic signature of monsoon conditions, cloud modes, and rainfall type. Hydrological Processes. 2018;32:2296–2303. https://doi.org/10.1002/hyp.13140

---

## Referee Comment (RC2) · Anonymous Referee #2 · 16 Oct 2018

In this manuscript the authors present an implementation of the heavy stable water isotope HDO in the National Taiwan University (NTU) microphysical scheme of the Weather Research and Forecasting (WRF) model. Isotopic fractionation during both liquid and ice cloud formation is computed according to kinetic mass transfer principles, which is possible because the NTU scheme does not apply the saturation adjustment technique. The model was subsequently used to study a cold front event in northern Taiwan, and disentangle different mechanisms that led to a decrease of $\delta$D in vapor and precipitation. The results show that cloud microphysical processes, the initial vertical distribution of $\delta$D, and lower boundary conditions are all similarly important for reproducing the observed evolution of $\delta$D during the cold front passage.

[Figure]

This is a well-written and interesting manuscript that takes advantage of the kinetic mass transfer formulation in the NTU scheme to physically simulate isotopic fractionation during cloud formation, avoiding equilibrium assumptions or uncertain parameterizations of supersaturation. The study nicely demonstrates how numerical models can help interpret isotope measurements and improve our understanding of the water cycle also on short time scales. I recommend that the manuscript be published after minor revisions.

General comments

1) The bibliography could be updated with some newer references. For example there are now isotope models that no longer apply saturation adjustment during ice cloud formation, e.g., IsoSAM (Blossey et al. 2010), COSMOiso (Pfahl et al., 2012).

2) As per comment (1) it would be interesting to see an additional experiment (maybe in the supplement) that assumes thermodynamic equilibrium only for the liquid phase, but not for ice. The nonequilibrium effect is generally assumed to be much stronger during ice than during liquid cloud formation due to higher supersaturations. Such an experiment would show how accurate this assumption is, and whether applying saturation adjustment and isotopic equilibrium during liquid cloud formation (as done in the models mentioned above) is reasonable or not.

3) Due to the low diffusivities of water molecules in ice there is no homogenization of isotopes in ice crystals and snow flakes. During deposition, the vapor only "sees" the outermost layer of the ice crystal / snow flake. Could you add a few sentences on how you define the isotopic composition of this outermost layer in the NTU scheme? For example, is it equal to the bulk composition of ice / snow, or the composition of the deposition flux? What is your rationale for one or the other?

4) Supersaturation and associated nonequilibrium effects are especially important for the second-order isotope parameter deuterium excess (d=$\delta$D - 8*$\delta$18O) . It would be interesting to see the results of this case study also for deuterium excess, which would,

however, require having H218O in the model. This could be something for a future study.

5) There are some minor English mistakes / typos, which I won't correct here. I suggest to ask someone for proofreading before resubmission.

Specific comments

Line 63: Better "liquid or ice and gas phases". For ice, supersaturation is (presumably) even more important than for liquid.

Line 195: I personally don't like the factor 1000 in the definition of $\delta$, because it is not part of the definition but part of how the definition is expressed. A similar thing would be to define pressure as p = F/A * 0.01 because it is often expressed in hPa. But this is a detail.

Line 207: "the ratio between the HDO concentration (QIV) and QV changes rather linearly with height." I am confused about this sentence. It does not seem linear in Fig. 4b.

Line 260-262: This sentence is not very clear. Do you mean depletion due to rain-out of heavy isotopes? Strong fractionation would otherwise lead to higher $\delta$D in the hydrometeors.

Line 283: Better "low $\delta$DV" instead of "significant $\delta$DV".

Line 304: Do you mean $\delta$DI (instead of $\delta$DV)?

Line 322: Better "the decrease in $\delta$DV was overestimated", because $\delta$DV itself is underestimated.

Line 328: For consistency I would also discuss the NoFrac simulation already in the results section.

Line 370: Reference for precipitation measurements?

Line 380: Also evaporation from the ocean can have a large influence on the isotope values close to the surface. As far as I understood, this is not explicitly considered in the model. Is that correct?

Line 403: "overestimate the decrease of $\delta$D", or "underestimate $\delta$DV".

Line 410: Same as line 403 but the other way round.

Fig. 1: Write out the terms CCN, GCCN, IN in the figure caption.

Fig. 4b: Labels for the axes would be helpful.

Fig. 6: Is this a daily average? Please specify.

Fig. 7: For consistency "observed (OBS: black line)". Rotate time axis labels so they are more readable.

Fig. 10: Figure resolution is bad, labels are small and not readable. Where do you calculate the vertical distribution? Is the x axis in space or time?

Fig. 12: Add units.

References

Blossey, P. N., Z. Kuang, and D. M. Romps (2010). Isotopic composition of water in the tropical tropopause layer in cloud-resolving simulations of an idealized tropical circulation. J. Geophys. Res., 115, D24309, doi:10.1029/2010JD014554

Pfahl, S., H. Wernli, and K. Yoshimura (2012). The isotopic composition of precipitation from a winter storm – a case study with the limited-area model COSMOiso. Atmos. Chem. Phys., 12, 1629–1648, doi:10.5194/acp-12-1629-2012

---

## Author Comment (AC1) · 27 Nov 2018

**Response to Anonymous Referee #1**

**Summary**

Tsai et al. have made a case study of kinetic isotope effects in clouds during a precipitation event over Taiwan. The authors have implemented two isotopologues of water in the hydrological cycle of a regional model for this purpose. The study uses a growth model for the distribution of the droplet size in the cloud scheme. The conclusion is that kinetic effects have a significant effect on the isotope composition, and including these effects yields a result closer to observations.

**Major Comments**

More studies of cloud process using high-resolution models are most certainly needed to further our understanding, both of the cloud processes themselves, as well as for understanding the isotope fractionation processes. While this study attempts to do these things I think that the manuscript in the current form only got the authors half way there.

1) I think the premises for this manuscript are not correct, or at least very imprecise. As I outline in the comments below (e.g. comment for L62-64) the authors use a 20 year old paper to motivate their study and generalize the research area. Quite some major studies have been published in the meantime. While the results by Tsai et al. might be correct and their model well functioning, the reader has little chance to know what is actually new in this study. Furthermore, it appears to me that the manuscript in its present form might lead the reader to think that the approach of the study is more novel than it actually is (comments to L62-64 and L128).

**Reply:** Thanks for pointing out our deficiency in reviewing current progress in isotope models. We have updated the manuscript with more current studies in section 1 as the following: "In conventional AGCMs, isotope exchange between liquid or ice and gas phases is usually assumed to be in a partial or full equilibrium state [*Hoffmann et al.*, 1998, Risi et al., 2010, Nusbaumer et al., 2017, Werner et al., 2011, Yoshimura et al., 2010]. In a synoptic weather system such as a front or typhoon, thermal equilibrium fractionation may not be appropriate for describing fractionation during phase change since the clouds are usually not in vapor equilibrium [ *Laskar et al.*, 2014]. Therefore, in recent years, several regional models start to consider kinetic fractionation during evaporation from open water, condensation from vapor to ice, or isotope exchange from raindrops to unsaturated air [Hoffmann et al., 1998, Yoshimura et al., 2010; Blossey et al. 2010; Pfahl et al., 2012; *Dütsch et al.*, 2016]."We also described the advantages of our scheme more clearly in section 2.1 with the following: "The NTU scheme is a two-moment scheme that predicts both the number and mass concentrations of each bulkwater category, which allows better presentation of microphysical processes than the commonly used one-moment schemes [*Taufour et al.*, 2018]. In contrast to the conventional bulkwater schemes that must assume a certain size distribution function, the NTU scheme derived the warm-cloud parameterization by analyzing results from bin model simulations and thus is rather accurate and comprehensive in microphysical processes; while the cold-cloud parameterization still follows the conventional approach. Another advantages of the NTU scheme is that it does not apply the "saturation adjustment" strategy, as done in most global and regional models. This saturation adjustment treatment assumes that water vapor and liquid (or ice) water

are in thermodynamic equilibrium once water (or ice) saturation is reached in non-mixed-phase clouds (i.e., all hydrometeors are either liquid or ice). In mixed-phase clouds (i.e., water and ice coexist), the equilibrium is maintained by varying linearly from water saturation to ice saturation between two specified temperature thresholds. Then, condensation on ice can be calculated following the kinetic approach, but the condensation on cloud drops still follows the saturation adjustment in most models. If the air is subsaturated but with the presence of cloud drops (or cloud ice), the cloud drops (or cloud ice) are forced to evaporate to maintain the equilibrium until they are all evaporated. Therefore, under the saturation adjustment assumption, kinetic effect as described in Eq. (5) cannot be solved fully and explicitly.

However, the kinetic effect might have significant impacts on isotope fractuonation and thus needs to be considered in models. For example, *Hoffmann et al.* [1998] tried to consider the kinetic effect during deposition growth in the ECHAM AGCM model. Due to the saturation adjustment assumption in ECHAM model, an effective factor, which is function of temperature only, is used to express the kinetic effect [*Jouzel and Merlivat*, 1984]. In *Wernet et al.* [2011], the condensation on ice is also calculated with an effective factor, but the condensation on cloud drops is in equilibrium fractionation. In reality, deviation from equilibrium is rather common in cloud, and its magnitude depends on factors such as updraft speed and hydrometeors' size spectra. These factors usually are not considered in existing models, but are included in the NTU scheme. Note that the saturation adjustment strategy conventionally is not applied in subsaturated conditions for precipitation particles (e.g., raindrops, snow, etc.), so the it should be denoted as a partial equilibrium assumption."

2) Basic analysis of the relationship between precipitation and $\delta$D is missing. As I understand it, this isotope enabled version of the WRF model has not previously been published. For a new model I would expect more extensive validation. At least for the coarse-resolution model simulation. For the larger domain, much more data are available for a general evaluation. Then the detailed case study comes after, if the model shows reasonable performance. The authors mentions the amount effect once in the introduction never to return to it, neither in the analysis nor in the discussion. I suggest Kurita (2013) and Zwart et al. (2018) as a starting point. I also suggest a few more plots (see last two comments for Figures) that would be very helpful for the analysis and for the reader to have some fundamental understanding of the performance of the model.

**Reply:** The evaluations of cloud microphysical processes, which were not affected by isotope fractionation, have been provided by Cheng et al. (2010), Chen et al. (2015), and Dearden et al. (2016). For water stable isotopes, the NASA Tropospheric Emission Spectrometer (TES) on board the Aura satellite provides the best observation for regional verification. However, for the dates that simulated, Aura satellite's tracks were over the North America (Fig. R1) only monthly mean data , which is used as initial condition, is available (see figures below). Therefore, we rely on the surface measurements of water stable isotope in the vapor phase and precipitation (Fig. 7 in the revised manuscript), conducted in Taiwan for verification. Overall, the model captured reasonably well the pattern and magnitude of changes

in δD during the frontal passage, except that the timing is off by a few hours. As for the amount effect, we have included Kurita (2013) and Zwart et al. (2018) in the reference, and also added the following sentences for a brief description: "The precipitation amount effect states that isotopic contents of tropical precipitation decrease as the amount of local precipitation increases [*Dansgaard*, 1964; *Kurita*, 2013], and the cause of which could be either the preferential removal during condensation [*Cole et al.*, 1999; *Yoshimura et al.*, 2003] or stronger downdraft in more intense convection [*Risi et al.*, 2008]."

[Figure]

Figure R1: TES measurement tracks on June 11-13, 2012.

3) When it comes to the writing, formulations are often not precise enough when describing specific processes (e.g. comments to L62-64, L86, L88, L128, L177-178, L259-260, L332-334).
**Reply:** We revised the descriptions and formulations as described in specific comments.

4) The quality of plots and labels make it difficult for the reader to see the point the authors are trying to make. See comments for Figures.
**Reply:** We revised the figures as described in comments for figures.

Given these major comments and the specific comments below I cannot recommend this study for publication in its current form. I think it would be better to resubmit after recasting the study. I hope that the authors will take the time to do this, and maybe an updated study could be an interesting contribution to the topic of water isotopes in the climate system.

**Specific comments**
L18 Insert "obtain" after "to".
**Reply:** Revised accordingly.

L22 The authors are careful to use the term "isotopocule", but the title says "water isotope". The colloquial, but technically incorrect, expression "water isotopes" is widely used. Whatever the authors choose, please be consistent in terminology. Generally, "isotopologue" is more widely used for water isotopes than "isotopocule", since they for water mean the same thing. A less heavy, but still correct, term would be "water stable isotopes".
**Reply:** Thanks for the suggestion. We have used "water stable isotopes" wherever possible in the revised manuscript.

L31 Replace "often" with "generally" and replace "variation" with "variations". L32

Replace "location" with "space".
**Reply:** Revised accordingly.

L36-43 It would be helpful to the reader to add specific references for specific processes discussed.
**Reply:** Thanks for the suggestion.  We revised the description as: "These factors are related to various physical processes, such as the surface water vapor source, atmospheric transport, phase changes in clouds and gravitational sorting of precipitation hydrometeors. For example, the water stable isotopic ratios decreased inland from the coast and the so-called continental effect [*Clark and Fritz*, 1997]. The precipitation amount effect states that isotopic contents of tropical precipitation decrease as the amount of local precipitation increases [*Dansgaard*, 1964; *Kurita*, 2013], and the cause of which could be either the preferential removal during condensation [*Cole et al.*, 1999; *Yoshimura et al.*, 2003] or stronger downdraft in more intense convection [*Risi et al.*, 2008]."

L62-64 Please be more precise when describing the what you mean by "partial or full equilibrium state". In Hoffmann et al. (1998) kinetic fractionation is taken into account during evaporation and snow formation, as well as partial evaporation of raindrops when the air is undersaturated. Kinetic isotope effects have been investigated in a number of GCM studies, here among: Schmidt et al. (2005), Risi et al. (2010), Werner et al. (2011) and Nusbaumer et al. (2017). Given the motivation and focus of the study, the authors owe it to the reader to highlight the existing literature and provide a precise description to elucidate what is novel in this study. Could it be that the authors are also not familiar with Yoshimura et al. (2010)? I would surely expect such a paper to be cited for this topic.
**Reply:** Yes, more precise explanation of "partial or full equilibrium state" is needed, and has been added in Section 2 (see reply to Major comment #1).

L72 Replace "Duterium" with "Deuterium"?
**Reply:** Revised accordingly.

L72-74 Please reformulate. This means you only incorporate HDO and $H_2^{16}O$? And if so, why? Kinetic effects have strong impact on the deuterium excess.
**Reply:**  We added the reason in section 1 as: "Because the $\alpha_{l-v}$ of $^{18}O$ (grey line in Fig. 2) does not deviate significantly from unity, so the signal of $^{18}O$ fractionation is generally much less pronounced. Therefore, we focus on deuterium for demonstrating the fractionation processes."

L86 With "saturation adjustment" do you refer to parameterizations of supersaturation? E.g. where supersaturation is a linear function of temperature below a threshold?  Please provide an example and reference.
**Reply:** We have added explanations in section 2; see reply to major comment #1 for details.

L88 "according to kinetic mass transfer principles" so this is not a parameterization, but explicit physics?

**Reply:** The condensation/evaporation processes are calculated explicitly according to Eq. (5), except that the NTU scheme provided a parameterization for integrating Eq. (5) over the whole drop size spectrum. We have revised the manuscript accordingly.

L98 I assume $H_2O$ means $H_2^{16}O$? Please specify.
**Reply:** Thanks for pointing it out. We modified the sentence as: "the HDO concentration can be determined from the $H_2^{16}O$ (hereafter, $H_2O$)"

L119-122 Have you tested the sensitivity to these formulations? In most models the formulation for ice/vapor by Merlivat and Nief (1967) is still used.
**Reply:** The comparison of $\alpha$ between this study and Merlivat and Nief (1967) is shown Fig. 2. We added relevant discussion in section 4 as: "Another uncertainty is the parameterization of isotopic fractionation factor $\alpha$. In this study, the temperature dependence of $\alpha_{l-v}$ was from *Horita and Wesolowski* [1994] and that between ice and water vapor $\alpha_{s-v}$ was adapted from *Ellehoj et al.* [2013]. In most models, the formulation for ice/vapor by *Merlivat and Nief* (1967) is still used. The differences in $\alpha_{s-v}$ between *Ellehoj et al.* [2013] and *Merlivat and Nief* (1967) are around 1% between -10~-20°C and 4% at -40°C. The differences of $\alpha_{l-v}$ between *Horita and Wesolowski* [1994] and *Merlivat and Nief* (1967) are less than 1%. "

L128 As also noted by Robinson and Scott (1981) this equation is derived using similar arguments as Fick's law, and essentially says that the flux is proportional to the gradient of the concentration. The formulation of kinetic fractionation for bulk vapor is derived using Fick's law (e.g. Merlivat and Jouzel, 1984), and thus very similar to what is done in Eq. 5. My point being, that the real difference from your study to other isotope model studies is not that you take kinetic fractionation into account and they don't, because they actually do (see comment to L62-64). The difference is in the widely used bulkwater formulation, while you use a growth model for the distribution of the droplet size, and formulate the isotope fractionation accordingly (following existing principles). Giving the reader something to hold on to in comparing your work to previous work (e.g. Eq. 5 following similar principles as Fick's law, which other models base their kinetic isotope scheme on) is important to convey where your study is placed compared to other studies.
**Reply:** This question is a re-emphasis of comment #1, and we thank you for pointing out this blind spot. We have made it more clear in the revised section 2.1. See reply to comment #1 for details.

L145-147 Is this the motivation for the text L72-74?
**Reply:** Yes, we moved the sentences to section 1 to make it more clear.

L145 In Fig. 2, can you provide a comparison with liquid/solid phase vapor pressure using the more common parameterizations found in GCMs?
**Reply:** Thanks for the suggestions. We have modified Fig.2 and added the discussion in section 4. See earlier reply to comment on L119-122.

L162 What about spin up of simulations? Longer spin up is usually required for isotope simulations.

**Reply:** The spin up of simulations is 12 hours in this study. The treatments of initial and boundary conditions as discussed in section 2.2 helps to reduce the spin up time.

L177-178 How is the nudging done? Spectral nudging or?
**Reply:** In this study, analysis-nudging is used every 6 h for domains 1 and 2 only. We have added these model details in section 2.

L197-215 So the observed d$^{18}$O of sea water (e.g. LeGrande and Schmidt, 2006) is not used as lower boundary conditions, and your vapor isotope boundary conditions are derived from mean conditions? Sturm et al. (2005) showed that regional simulations are very sensitive to isotope boundary conditions, obtaining the best results with a nudged global model run as boundary conditions. Please provide some arguments how you can use this type of boundary conditions for a case study. For example, if the variability generally is small (please quantify) then it could be argued that, an observed mean "climatology" is representative. I am aware that you partly test the boundary conditions with the "NoLnd" run, but this is not what I'm asking for.
**Reply:** The observed mean climatology in June from GNIP was used as the lower boundary condition in this study. We revised the sentence as: "The lower boundary condition of δD over land and ocean are calculated by relating HDO flux to H$_2$O flux according to Eqs. (3) and (4). In such as conversion, the ratio $R_l$ is set to be that in surface precipitation according to observed mean climatology in June from the Global Network of Isotopes in Precipitation (GNIP)".

L234-236 That the model captures the location front is no surprise due to the nudging. Or is the nudging not constraining the model very tightly?
**Reply:** As mentioned in section 2, FNL data for wind properties and temperatures were nudged into domains 1 and 2 but not in domain 3. Because we do not want the nudging practice to influence cloud microphysics in our focused area.

L259-260 You need to explain the continental effect somewhere, as you use this argument several times. This could for example be done in the introduction where the classic isotope effects are only touched upon in very general fashion.
**Reply:** Thanks for the suggestion. We added some description in section 1 to explain the continental effect "The isotopic ratios decreased inland from the coast and the so-called continental effect [*Clark and Fritz,* 1997]."

L267 Replace "mechanimss" with "mechanisms".
**Reply:** Revised accordingly.

L285-287 Please reformulate. Maybe simply replace "discrepancy" with "biases". Also, hasn't several studies already shown that this is generally true (e.g. Risi et al., 2010)? Does your study have smaller biases than other studies?
**Reply:** Agree. We have replaced "discrepancy" with "biases" in the revised manuscript, and also mentioned that this finding is similar to other studies such as [Risi et al., 2010].

L288-291 This is one explanation, which sounds like an instantaneous process no matter the history of the air parcel. What about the progressive rain out of heavy isotopologues during adiabatic ascend?

**Reply:** Sorry for the confusion. Line 288-291 is motivation, not explanation, of NoIce run. The effect of precipitation is discussed in section 3.2 as: "Secondly, precipitation inside the frontal system caused a strong reduction (fractionation) in $\delta D$ of hydrometeors as can be seen in Figs. 8e and 8f; therefore, the evaporation of hydrometeors would produce low $\delta D_v$ in the lower troposphere."

L295 Replace "." with white space?
**Reply:** Revised accordingly.

L297 Replace "heavier" with "heavy".
**Reply:** Revised accordingly.

L332-334 Do you mean that there is a local marine source at the onset of the precipitation? In this context what does microphysical processes? I don't follow what the authors mean in this paragraph. Please rephrase.

**Reply:** From the surface weather map and sounding, there were strong southwesterly flows on 11 June, which indicated the water vapor source of the precipitation is from the ocean (Wang et al., 2016). We modified the first paragraph in section 4 as: "Combining the observations and simulations results of the stable water isotopes can be used to understand the water cycle. From the observed $\delta D_v$ decreased after 06:00 on 12 June (black line in Fig. 11), much later than the onset of the precipitation. The observations suggested that the source of water vapor before this time is the ocean [Wang et al., 2016], and that the microphysical processes related to the precipitation did not substantially affect $\delta D_v$ during this period."

L340 Replace "mid night" with "midnight".
**Reply:** Revised accordingly.

Figures:
* Please add axis labels and units to all graphs. E.g. Figure 4b has neither axis labels nor units, also no mention of units in the caption.
**Reply:** Revised accordingly.
* Labels and axis on Figure 9 and 10 are next to impossible to read. Please provide readable plots.
**Reply:** Revised accordingly.
* It is very difficult to see the differences between the curves (especially Figure 7 and 11) if you want to assess the discrepancy in time between the different curves. I suggest
 i) place subplots in upper and lower panels instead of side-by-side to "stretch" time
ii) add axis grid in plot
iii) make more readable axis labels that don't line up so close, for example by rotating them 45 degrees, and make it clear what is date and what is time.
**Reply:** Thanks for the suggestions. They are revised accordingly.
* It would be extremely helpful to plot $\delta D$ and precipitation rates in the same plot

to see if the timing of changes in $\delta D$ and precipitation amount is similar in model and observations.

* I suggest that the authors also add plots of $\delta D$ vs precipitation and compare the model to observation and assess the classic amount effect. This is featured in many studies of isotopes in precipitation at low latitudes, and would give a way to compare to other studies.

**Reply:** Fig. R2 is the time series of $\delta D$ and precipitation rates (same as Fig. 5b and 7 in the revised manuscript). We added some description in the last paragraph of section 3.1 "The classic amount effect cannot be assessed from observations. For model simulations, the simulated $\delta D$ in precipitation (Fig. 7b) decreased with precipitation occurred (Fig. 5b). The negative correlation is similar to the amount effect in other studies. Overall, the model captured reasonably well the pattern and magnitude of changes in $\delta D$ during the frontal passage, except that the timing is off by a few hours."

[Figure]

Figure R2. Simulated (CTRL: red line) and observed (OBS: black line) water vapor $\delta D_v$ (upper panel, in ‰) at AS and precipitation $\delta D_p$ (middle panel, in ‰) at NTU. The bottom panel is simulated (red line) and observed (black line) precipitation (mm/hr) at Taipei station on 11-13 June 2012.

References

Chen, J. P., and S. T. Liu (2004), Physically based two-moment bulkwater parametrization for warm-cloud microphysics, Quarterly Journal of the Royal Meteorological Society, 130(596), 51-78, doi:10.1256/qj.03.41.

Ellehoj, M., H. C. Steen-Larsen, S. J. Johnsen, and M. B. Madsen (2013), Ice-vapor

equilibrium fractionation factor of hydrogen and oxygen isotopes: Experimental investigations and implications for stable water isotope studies, *Rapid Communications in Mass Spectrometry*, *27*(19), 2149-2158.

Horita, J., and D. J. Wesolowski (1994), Liquid-vapor fractionation of oxygen and hydrogen isotopes of water from the freezing to the critical temperature, *Geochimica et Cosmochimica Acta*, *58*(16), 3425-3437.

Kurita, N. (2013), Water isotopic variability in response to mesoscale convective system over the tropical ocean, J. Geophys. Res. Atmos., 118, doi:10.1002/jgrd.50754.

LeGrande, A. N., and G. A. Schmidt (2006), Global gridded data set of the oxygen isotopic composition in seawater, Geophys. Res. Lett., 33, L12604, doi:10.1029/2006GL026011.

Merlivat L, and Nief G (1967), Fractionnement isotopique lors des changements d'etat ´ solide-vapeur et liquide-vapeur de l'eau a des temp ` eratures inf ´ erieures ´ a 0 ` ∘C. Tellus 19:122–127.

Nusbaumer, J., T. E. Wong, C. Bardeen, and D. Noone (2017), Evaluating hydrological processes in the Community Atmosphere Model Version 5 (CAM5) using stable isotope ratios of water, J. Adv. Model. Earth Syst., 9, 949–977, doi:10.1002/2016MS000839.

Risi, C., S. Bony, F. Vimeux, and J. Jouzel (2010), Water stable isotopes in the LMDZ4 general circulation model: Model evaluation for present day and past climates and applications to climatic interpretations of tropical isotopic records, J. Geophys. Res., 115, D12118, doi:10.1029/2009JD013255.

Robinson, N.F., Scott, W.T., 1981. Two-stream Maxwellian kinetic theory of cloud droplet growth by condensation. J. Atmos. Sci. 38, 1015–1026.

Schmidt, G. A., G. Hoffmann, D. T. Shindell, and Y. Hu (2005), Modeling atmospheric stable water isotopes and the potential for constraining cloud processes and stratosphere-troposphere water exchange, J. Geophys. Res., 110, D21314, doi:10.1029/2005JD005790.

Sturm, K., G. Hoffmann, B. Langmann, and W. Stichler (2005), Simulation of delta O-18 in precipitation by the regional circulation model REMOiso, Hydrol. Process., 19(17), 3425–3444, doi:10.1002/hyp.5979.

Werner, M., P. M. Langebroek, T. Carlsen, M. Herold, and G. Lohmann (2011), Stable water isotopes in the ECHAM5 general circulation model: Toward high-resolution isotope modeling on a global scale, J. Geophys. Res., 116, D15109, doi:10.1029/2011JD015681.

Yoshimura, K., M. Kanamitsu, and M. Dettinger (2010), Regional downscaling for stable water isotopes: A case study of an atmospheric river event, J. Geophys. Res., 115, D18114, doi:10.1029/2010JD014032.

Zwart C, Munksgaard NC, Protat A, Kurita N, Lambrinidis D, Bird MI. The isotopic signature of monsoon conditions, cloud modes, and rainfall type. Hydrological Processes. 2018;32:2296–2303. https://doi.org/10.1002/hyp.13140

---

## Author Comment (AC2) · 27 Nov 2018

**Response to Anonymous Referee #2**

In this manuscript the authors present an implementation of the heavy stable water isotope HDO in the National Taiwan University (NTU) microphysical scheme of the Weather Research and Forecasting (WRF) model. Isotopic fractionation during both liquid and ice cloud formation is computed according to kinetic mass transfer principles, which is possible because the NTU scheme does not apply the saturation adjustment technique. The model was subsequently used to study a cold front event in northern Taiwan, and disentangle different mechanisms that led to a decrease of $\delta$D in vapor and precipitation. The results show that cloud microphysical processes, the initial vertical distribution of $\delta$D, and lower boundary conditions are all similarly important for reproducing the observed evolution of $\delta$D during the cold front passage. This is a well-written and interesting manuscript that takes advantage of the kinetic mass transfer formulation in the NTU scheme to physically simulate isotopic fractionation during cloud formation, avoiding equilibrium assumptions or uncertain parameterizations of supersaturation. The study nicely demonstrates how numerical models can help interpret isotope measurements and improve our understanding of the water cycle also on short time scales. I recommend that the manuscript be published after minor revisions.

**General comments**

1)  The bibliography could be updated with some newer references. For example there are now isotope models that no longer apply saturation adjustment during ice cloud formation, e.g., IsoSAM (Blossey et al. 2010), COSMOiso (Pfahl et al., 2012).

**Reply:** Agree. We have revised the introduction section and added some newer references as suggested: "In conventional AGCMs, isotope exchange between liquid or ice and gas phases is usually assumed to be in a partial or full equilibrium state [*Hoffmann et al.*, 1998, *Risi et al.*, 2010, *Nusbaumer et al.*, 2017, *Werner et al.*, 2011, *Yoshimura et al.*, 2010]. In a synoptic weather system such as a front or typhoon, thermal equilibrium fractionation may not be appropriate for describing fractionation during phase change since the clouds are usually not in vapor equilibrium [ *Laskar et al.*, 2014]. Therefore, in recent years, several regional models start to consider kinetic fractionation during evaporation from open water, condensation from vapor to ice, or isotope exchange from raindrops to unsaturated air [*Hoffmann et al.*, 1998, *Yoshimura et al.*, 2010; *Blossey et al.* 2010; *Pfahl et al.*, 2012; *Dütsch et al.*, 2016]. However, the microphysics in these global or regional models are usually described with single moment schemes.".  We also add more description about the concept of "saturation adjustment" in section 2.1 in the revised manuscript: "Another advantages of the NTU scheme is that it does not apply the "saturation adjustment" strategy, as done in most global and regional models. This saturation adjustment treatment assumes that water vapor and liquid (or ice) water are in thermodynamic equilibrium once water (or ice) saturation is reached in non-mixed-phase clouds (i.e., all hydrometeors are either liquid or ice). Therefore, for models applying the saturation adjustment strategy, condensation is not calculated explicitly but rather by converting all excess water vapor into condensate regardless of the cloud drop size and number concentration or the time needed for condensing out all supersaturated water. So, under the saturation adjustment assumption, kinetic effect as described in Eq. (5) cannot be solved fully and explicitly. In mixed-phase clouds (i.e., water and ice coexist), the equilibrium is maintained by assuming either water saturation or ice saturation (e.g., *Sundqvist*, 1978), or by varying linearly from water saturation to ice

saturation between two specified temperature thresholds (e.g., *Tiedtke*, 1993). Then, condensation on ice can be calculated following the kinetic approach, but the condensation on cloud drops still follows the saturation adjustment in most models. If the air is subsaturated but with the presence of cloud drops (or cloud ice), the cloud drops (or cloud ice) are forced to evaporate to maintain the equilibrium until they are all evaporated. As the saturation adjustment strategy conventionally is not applied in subsaturated conditions for precipitation particles (e.g., raindrops, snow, etc.), it should be denoted as a partial equilibrium assumption.".

2) As per comment (1) it would be interesting to see an additional experiment (maybe in the supplement) that assumes thermodynamic equilibrium only for the liquid phase, but not for ice. The nonequilibrium effect is generally assumed to be much stronger during ice than during liquid cloud formation due to higher supersaturations. Such an experiment would show how accurate this assumption is, and whether applying saturation adjustment and isotopic equilibrium during liquid cloud formation (as done in the models mentioned above) is reasonable or not.

**Reply:** Thanks for the nice suggestion. We added an extra simulation assuming thermodynamic equilibrium for the liquid phase only (LiqEQ). As shown in Fig. R1. the decrease in $\delta D_v$ was reduced by 20-50‰ in the LiqEQ run before the frontal passage in the early morning of 12 June, but enhanced by 10‰ at one point of frontal passage (06:00-12:00 LST on June 12). As shown in Figure R1b, the decrease in $\delta D_l$ was enhanced when the saturation adjustment treatment was applied. But we would expect that this "partial equilibrium" simulation would produce results sitting somewhere between those of the CTRL and EQ runs. So, we are not very confident of including these new results. Due to the interest of time, we would like to leave it out of this revision. In the mean time, we will re-check/re-run the simulations and see if something more concrete can be obtained for the next revision (if allowed).

[Figure]

Figure R1. (a) Same as Fig. 11a. Simulated (CTRL: red line) and observed (OBS: black line) water vapor $\delta Dv$ (in ‰) at "AS" on 11-13 June 2012; and (b) Difference in simulated $\delta D$ (in ‰) of water vapor (left) and liquid-phase condensates (right) at 850 hPa between CTRL and LiqEQ on 12 June, 2012.

Due to the low diffusivities of water molecules in ice there is no homogenization of isotopes in ice crystals and snow flakes. During deposition, the vapor only "sees" the outermost layer of the ice crystal / snow flake. Could you add a few sentences on how you define the isotopic composition of this outermost layer in the NTU scheme? For example, is it equal to the bulk composition of ice / snow, or the composition of the deposition flux? What is your rationale for one or the other?

**Reply:** Thanks for the suggestion. We added the following descriptions: "In Eq. (6), the activity of water stable isotope depends on the composition of the particle. For ice particles, the model cannot trace the history of water stable isotope deposition and thus cannot distinguish between the surface layer from the inner core of the ice particles. Therefore, the water stable isotope activity of ice-phase hydrometeor is assumed to depend on its bulk composition (i.e. assuming well-mixed). In reality, however, there is no homogenization of isotopes in ice particles due to the low diffusivities of molecules in ice. *Blossey et al.* [2010], *Pfahl et al.* [2012] and *Dütsch et al.* [2016] dealt with this problem by setting the ice particle's isotope ratio equal to that produced by vapor deposition. This is an effective approach as only the most recently deposited ice is exposed to the vapor. However, during evaporation the mass

exchange depends heavily on the residual composition, making the treatment rather tricky. Before a better solution is devised, this study adopted the bulk composition approach for both condensation and evaporation processes." We also mentioned a suggested approach in the conclusion section as the following: "The problem in determining the activity of water stable isotope in ice particles without knowing the inhomogeneity of chemical composition in the bulk ice, as mentioned at the end of section 2.1 is another issue worthy of further study. To accommodate the different conditions between condensation and evaporation, it might be feasible to assume that the water stable isotope activity is determined by the vapor phase during condensation following the approach of *Blossey et al.* [2010], *Pfahl et al.* [2012] and *Dütsch et al.* [2016]; whereas for the evaporation process, one may assume a well-mixed bulk composition for determining the isotope activity as done in this study."

3) Supersaturation and associated nonequilibrium effects are especially important for the second-order isotope parameter deuterium excess (d=$\delta$D - 8*$\delta^{18}$O). It would be interesting to see the results of this case study also for deuterium excess, which would, however, require having $H_2^{18}$O in the model. This could be something for a future study.

**Reply:** Deuterium was selected because we expect that the fractionation effect for $H_2^{18}$O would be less significant according to Fig. 2. But, we do agree that it will be interesting to test it out in the future. This is mentioned in the discussion section as "Finally, whether the nonequilibrium effects are important for the second-order isotope parameter, deuterium excess, is an interesting subject worthy of further investigation by including the description of $\delta^{18}$O isotope in the model."

4) There are some minor English mistakes / typos, which I won't correct here. I suggest to ask someone for proofreading before resubmission.

**Reply:** Thanks for the suggestion. We have asked help for proofreading in the revision.

**Specific comments**
Line 63: Better "liquid or ice and gas phases". For ice, supersaturation is (presumably) even more important than for liquid.
**Reply:** Agree. Revised accordingly.

Line 195: I personally don't like the factor 1000 in the definition of $\delta$, because it is not part of the definition but part of how the definition is expressed. A similar thing would be to define pressure as p = F/A * 0.01 because it is often expressed in hPa. But this is a detail.
**Reply:** Agree. We have remove the factor "1000" from Eq. (9).

Line 207: "the ratio between the HDO concentration (QIV) and QV changes rather linearly with height." I am confused about this sentence. It does not seem linear in Fig. 4b.
**Reply:** The absolute values of QIV and QV vary exponentially with height, but the ratio QIV:QV is quite linearly varying (which cannot be seen in Fig. 4b). We have modified the sentence as: "Although the concentrations of water vapor (QV) and HDO (QIV) usually decrease exponentially with height, their ratios (i.e., QV:QIV) vary rather

linearly with height." Hope it is less confusing now.

Line 260-262: This sentence is not very clear. Do you mean depletion due to rainout of heavy isotopes? Strong fractionation would otherwise lead to higher $\delta D$ in the hydrometeors.

**Reply:** Yes, the low $\delta D$ around the frontal system is due to precipitation. We modified the sentence as: "Secondly, precipitation inside the frontal system caused a strong reduction (fractionation) in $\delta D$ of hydrometeors as can be seen in Figs. 8e and 8f"

Line 283: Better "low $\delta DV$" instead of "significant $\delta DV$".
**Reply:** Revised accordingly.

Line 304: Do you mean $\delta DI$ (instead of $\delta DV$)?
**Reply:** Thanks for the correction. It is revised accordingly.

Line 322: Better "the decrease in $\delta DV$ was overestimated", because $\delta DV$ itself is underestimated.
**Reply:** Revised accordingly.

Line 328: For consistency I would also discuss the NoFrac simulation already in the results section.
**Reply:** Agree. We have added a brief description of the NoFrac result in the results section.

Line 370: Reference for precipitation measurements?
**Reply:** For precipitation measurements from the Central Weather Bureau of Taiwan, we added the website of CWB Taiwan in section 2.3 in the revised draft.

Line 380: Also evaporation from the ocean can have a large influence on the isotope values close to the surface. As far as I understood, this is not explicitly considered in the model. Is that correct?
**Reply:** This is correct. We added a few sentence in the last paragraph of discussion to clarify this: "In addition, the evaporation from the ocean is assumed as in equilibrium between liquid and vapor phases. This assumption may also affect the simulation of $\delta D$ in the model, and the process needs to be explicitly considered in the future.

Line 403: "overestimate the decrease of $\delta D$", or "underestimate $\delta DV$".
**Reply:** Revised accordingly.

Line 410: Same as line 403 but the other way round.
**Reply:** Revised accordingly.

Fig. 1: Write out the terms CCN, GCCN, IN in the figure caption.
**Reply:** Thanks for the suggestion. We modified the caption of Fig. 1 as: "Schematics of modified NTU scheme. The blue boxes are the hydrometeors considered in the model and the H/D indicated that both $H_2O$ and HDO are included. The arrows are the microphysical processes considered and the light blue boxes are cloud condensation

nuclei (CCN), giant CCN (GCCN) and ice nuclei (IN)."

Fig. 4b: Labels for the axes would be helpful.
**Reply:** Revised accordingly

Fig. 6: Is this a daily average? Please specify.
**Reply:** It is at 08:00 LST on 11 and 12 June.  We added the information in the caption.

Fig. 7: For consistency "observed (OBS: black line)". Rotate time axis labels so they are more readable.
**Reply:** Revised accordingly

Fig. 10: Figure resolution is bad, labels are small and not readable. Where do you calculate the vertical distribution? Is the x axis in space or time?
**Reply:** Figure quality is probably degraded during the conversion to pdf file, but we have revised the figures with larger labels.  The area over northern Taiwan is defined as (121-123˚E, 25-27˚N), and the x axis is time.  We have modified the figure caption to include these information.

Fig. 12: Add units.
**Reply:**  Revised accordingly

References
Blossey, P. N., Z. Kuang, and D. M. Romps (2010). Isotopic composition of water in the tropical tropopause layer in cloud-resolving simulations of an idealized tropical circulation. J. Geophys. Res., 115, D24309, doi:10.1029/2010JD014554
Pfahl, S., H. Wernli, and K. Yoshimura (2012). The isotopic composition of precipitationfrom a winter storm – a case study with the limited-area model COSMOiso. Atmos. Chem. Phys., 12, 1629–1648, doi:10.5194/acp-12-1629-2012
**Reply:**  Thanks for providing these useful references.